

# BAYESIAN ATMOSPHERIC CORRECTION OVER LAND: SENTINEL-2/MSI AND LANDSAT 8/OLI

Feng Yin[1,2], Philip E Lewis[1,2], and Jose L Gómez-Dans[1,2]

[1]Department of Geography, University College London, Gower Street, London WC1E 6BT, United Kingdom
[2]NERC National Centre for Earth Observation (NCEO)

**Correspondence:** Feng Yin (feng.yin.15@ucl.ac.uk)

**Abstract.**

Mitigating the impact of atmospheric effects on optical remote sensing data is critical for monitoring intrinsic land processes and developing Analysis Ready Data (ARD). This work develops an approach to this for the NERC NCEO medium resolution ARD Landsat 8 (L8) and Sentinel 2 (S2) products, called Sensor Invariant Atmospheric Correction (SIAC). The contribution of the work is to phrase and solve that problem within a probabilistic (Bayesian) framework for medium resolution multispectral sensors S2/MSI and L8/OLI and provide per-pixel uncertainty estimates traceable from assumed top-of-atmosphere (TOA) measurement uncertainty, making progress towards an important aspect of CEOS ARD target requirements.

A set of observational and *a priori* constraints are developed in SIAC to constrain an estimate of coarse resolution (500m) Aerosol Optical Thickness (AOT) and Total Column Water Vapour (TCWV), along with associated uncertainty. This is then used to estimate the medium resolution (10-60m) surface reflectance and uncertainty, given an assumed uncertainty of 5% in TOA reflectance. The coarse resolution *a priori* constraints used are: the MODIS MCD43 BRDF/Albedo product giving a constraint on 500m surface reflectance; and Copernicus Atmosphere Monitoring Service (CAMS) operational forecasts of AOT and TCWV providing estimates of atmospheric state at core 40 km spatial resolution with an associated 500m resolution spatial correlation model. The mapping in spatial scale between medium resolution observations and the coarser resolution constraints is achieved using a calibrated effective Point Spread Function for MCD43. Efficient statistical approximations (emulators) to outputs of the 6S atmospheric radiative transfer code used to estimate the state parameters and in the atmospheric correction.

SIAC is demonstrated for a set of global S2 and L8 images covering AERONET and RadCalNet sites. AOT retrievals show a very high correlation to AERONET estimates ($R$ around 0.86, RMSE of 0.07 for both sensors), although with a small bias in AOT. TCWV is accurately retrieved from both sensors ($R > 0.96$, $RMSE < 0.32 \ g/cm^2$). Comparisons with *in situ* surface reflectance measurements from the RadCalNet network show that SIAC provides accurate estimates of surface reflectance across the entire spectrum, with $RMSE$ mismatches with the reference data between 0.01 and 0.02 in units of reflectance, for both S2 and L8. For near-simultaneous S2 and L8 acquisitions, there is a very tight relationship ($R > 0.95$ for all common bands) between surface reflectance from both sensors, with negligible biases. Uncertainty estimates are assessed through discrepancy analysis and found to provide viable estimates for AOT and TCWV. For surface reflectance, they give conservative estimates of uncertainty, suggesting that a lower estimate of TOA reflectance uncertainty might be appropriate.



| Quantity | Threshold uncertainty | Reference |
|---|---|---|
| AOT | $0.05 + 0.15\mathrm{AOT}$ | Remer et al. (2009) |
| TCWV | $0.2 + 0.1\mathrm{TCWV}$ | Pflug et al. (2020) |
| $r$ | $0.005 + 0.05r$ | Vermote and Kotchenova (2008) |

Table 1: Threshold uncertainty specifications used in this paper.

## 1   Introduction

Land surface monitoring at optical wavelengths from medium resolution Earth Observation (EO) requires an accurate and consistent description of the bottom of atmosphere (BOA) spectral Bidirectional Reflectance Function (BRF) (Zhu et al., 2020) made readily available to users. This is acknowledged in efforts to develop consensus on 'analysis ready data' (ARD) (Wang et al., 2019; CEOS, 2019), and the value of such data for monitoring global change emphasised elsewhere (Feng et al., 2013; Hilker, 2018). Community needs for 'CEOS Analysis Ready Data for Land' (CARD4L) (CEOS, 2020) are stated as 'threshold' (minimum) and 'target' (desirable) requirements, with the former reflecting current practice and what is achievable with existing approaches and the latter an agreed position to move towards for the scientific and user communities. No uncertainty threshold or target values for surface reflectance are given in the CEOS ARD specification, but in this paper we use specifications adopted by Doxani et al. (2022) given in Table 1 as threshold values for Aerosol Optical Thickness (AOT), total column of water vapour (TCWV) and BOA BRF $r$ specification.

An important capability highlighted in target requirements is that per-pixel uncertainty estimates should be supplied (CEOS, 2021b), but this is lacking in the CEOS fully assessed USGS Landsat Collection 2 ARD product (CEOS, 2021a) and the ESA Sentinel-2 Level-2A (ESA, 2021b). Such information is vital for traceability and rigorous scientific analysis with ARD products (Merchant et al., 2017; Niro et al., 2021). Additionally, the ACIX intercomparisons of (Doxani et al., 2018, 2022) and surface reflectance comparison studies (Flood, 2017; Nie et al., 2019; Chen and Zhu, 2021) illustrate that further work is needed to ensure accuracy and consistency of such data, which is critical for combining data with different spatial, spectral, temporal and radiometric characteristics and achieving more comprehensive and/or frequent monitoring than with any single set (Lewis et al., 2012b; Wulder et al., 2015). This requires a focus on both accuracy and inter-operability that we suggest is not being adequately realised in current approaches.

In this paper, we describe the approach used for the UK NERC NCEO BOA BRF product from medium resolution S2/MSI and L8/OLI sensors (NCEO, 2021). It is sensor agnostic over the optical domain, in the sense that it does not rely on particular optical waveband sets, so is called Sensor Invariant Atmospheric Correction (SIAC). SIAC aims to be CARD4L-compliant at threshold requirements and move towards target requirements by providing per-pixel uncertainty values. This is enabled by applying a Bayesian framework to the estimation of atmospheric parameters from medium resolution multispectral observa-



tions and other constraints. The resulting parameters are used to derive an estimation of BOA BRF. Mean estimates derived from SIAC are validated against the criteria in Table 1 through global comparison of derived AOT and TCWV estimates with *in situ* AERONET measurements, comparisons of retrieved surface reflectance with *in situ* Radiometric Calibration Network (RadCalNet) measurements, as well as interoperability comparisons of surface reflectance between S2 and L8. The uncertainty

in the retrievals is also assessed, which complements the further validation of SIAC and inter-comparison with other processors in the ACIX-II experiment Doxani et al. (2022).

## 2 Atmospheric Correction Scheme in SIAC

### 2.1 Statement of the problem

We wish to estimate the probability distribution function (PDF) of BOA spectral BRF, $R$ with illumination and viewing vectors

$\Omega_s, \Omega_v$ respectively, on a grid $G_m$ of medium resolution pixels, over a set of wavebands $\Lambda_m$ (see Table 2 for symbol definitions). This is driven by medium resolution observations $Y$ under these conditions from S2/MSI and L8/OLI sensors at 10-60 m resolution, and other constraints. In SIAC as in most other approaches, we first seek an estimate of atmospheric state $X$ over the target scene. We assume multivariate Gaussian PDFs throughout, and ignore non-linear impacts on transformed distributions. Our approach targets the Maximum A Posteriori (MAP) estimate, given an *a priori* estimate of $X$, $X_b$ and the observations $Y$

mapped to a grid $G_c$ of coarse resolution pixels (nominal 500m resolution). We then apply $X$ at the original medium spatial resolution to the estimation of $R$ on $G_m$. The heritage of the approach is the various works on Bayesian/Optimal Estimation inference applied to mapping atmospheric parameters from EO for other sensors (Tanré et al., 2011; Dubovik et al., 2011; Lewis et al., 2012b; Dubovik et al., 2014; Govaerts and Luffarelli, 2018; Kaminski et al., 2017; Lipponen et al., 2018; Hou et al., 2020) as this gives the framework for combining multiple sources of information and estimating per-pixel uncertainty.

The MAP estimate of $X$ over $G_c$ is found by maximising the likelihood $P(X|Y)$ (Rodgers, 2000) :

$$P(X|Y) \propto P(Y|X)P(X) = \exp\left[-\mathcal{J}\right] \tag{1}$$

with $\mathcal{J} = \mathcal{J}_{obs} + \mathcal{J}_{prior}$. The mean MAP estimate of $X$ is achieved by minimising the negative logarithm of $P(X|Y)$ in Eq. (1), *i.e.* the 'cost function' $\mathcal{J}$ with respect to $X$. $X$ in the current version of SIAC contains AOT at 550 $nm$ and total TCWV in $g/cm^2$.

### 2.2 Overview

Our approach uses *a priori* constraints in the form of a coarse resolution (500m) spectral BRDF dataset from MODIS (Schaaf and Wang, 2015) to provide *sample* land surface reflectance estimates, as well as a very coarse resolution (40 km) estimate of atmospheric composition from the CAMS near-real-time global assimilation and forecasting system (Morcrette et al., 2009; Benedetti et al., 2009), with an associated sub-40 km spatial correlation constraint. These are combined with observational data

to solve an inverse problem to estimate atmospheric state at coarse resolution for the time and locations of the observations.



| Symbols | Meaning |
|---|---|
| $G_m$ | medium resolution grid of target sensor Level 1C data |
| $G_c$ | coarse resolution grid of atmospheric state variables |
| $d$ | relative day index, the day of a sample data point relative to the target day, $(-16 \leq d \leq 16)$ |
| $f_{iso}(d), f_{vol}(d), f_{geo}(d)$ | MDC43 BRDF model parameters for relative day $d$ on $G_c$ |
| $k_{vol}(\Omega_v, \Omega_s), k_{geo}(\Omega_v, \Omega_s)$ | MDC43 BRDF model kernels for angles $\Omega_v$ and $\Omega_s$ on $G_c$ |
| $\Lambda_m$ | set of native sensor wavebands of medium resolution sensor |
| $\Lambda_c$ | set of sensor wavebands of coarse resolution BRF |
| $D$ | first order spatial difference matrix, defined on $G_c$ |
| $R \sim \mathcal{N}(r, C_r)$ | *a posteriori* PDF of BOA spectral BRF defined over $\Lambda_m$ on $G_m$ |
| $R_b \sim \mathcal{N}(r_b, C_b)$ | *a priori* (background) PDF of BOA spectral BRF on $G_c$ over waveband set $\Lambda_m$ unless $\Lambda_c$ stated explicitly. Can also be specified as function of relative day $d$ as $R_b(d)$ |
| $X \sim \mathcal{N}(x, C_x)$ | PDF of atmospheric state variables defined on $G_c$ |
| $X_b \sim \mathcal{N}(x_b, C_{x_b})$ | *a priori* PDF of atmospheric state variables defined on $G_c$ |
| $Y \sim \mathcal{N}(y, C_y)$ | PDF of observations over $\Lambda_n$ defined on $G_m$ at $(\Omega_s, \Omega_v)$ |
| $Y_c \sim \mathcal{N}(y_c, C_{y_c})$ | PDF of observations over $\Lambda_m$ defined on $G_c$ at $(\Omega_s, \Omega_v)$ |
| $X_{a_c} \sim \mathcal{N}(x_{a_c}, C_{x_{a_c}})$ | augmented state vector containing $X$, BOA BRF estimate $R_b$ and ancillary variables (Ozone and altitude) defined on $G_c$ |
| $X_{a_m} \sim \mathcal{N}(x_{a_m}, C_{x_{a_m}})$ | augmented state vector containing $X$, TOA BRF $Y$ and ancillary variables (Ozone and altitude) defined on $G_m$ |
| $\hat{Y} \sim \mathcal{N}(\hat{y}, C_{\hat{y}})$ | PDF of modelled observations over $\Lambda_m$ defined on $G_c$ at $(\Omega_s, \Omega_v)$ |
| $\mathcal{J}_{obs}$ | observational negative log likelihood on $G_c$ |
| $\mathcal{J}_{prior}$ | *a priori* negative log likelihood on $G_c$ |
| $H(X_{a_c})$ | Observation operator $H$ that defines TOA spectral reflectance as a function of augmented state vector $X_{ac}$ |
| $\gamma$ | smoothness parameter used in differential constraint |
| $\bar{\bar{t}}^{\downarrow}$ | total (direct and diffuse) downwelling atmospheric transmittance, including modulation by gaseous absorption |
| $\bar{\bar{t}}^{\uparrow}$ | total (direct and diffuse) upwelling atmospheric transmittance |
| $\bar{\bar{r}}^{\downarrow}$ | spherical albedo of the atmosphere |
| $r^{\uparrow}$ | 'atmospheric intrinsic' or 'path' reflectance, i.e. the upwelling reflectance of the molecule and aerosol layer in direction $\Omega_v$ assuming a totally absorbing lower boundary, due to illumination from direction $\Omega_s$, modulated by gaseous absorption. |

Table 2: Main symbols used in the paper.



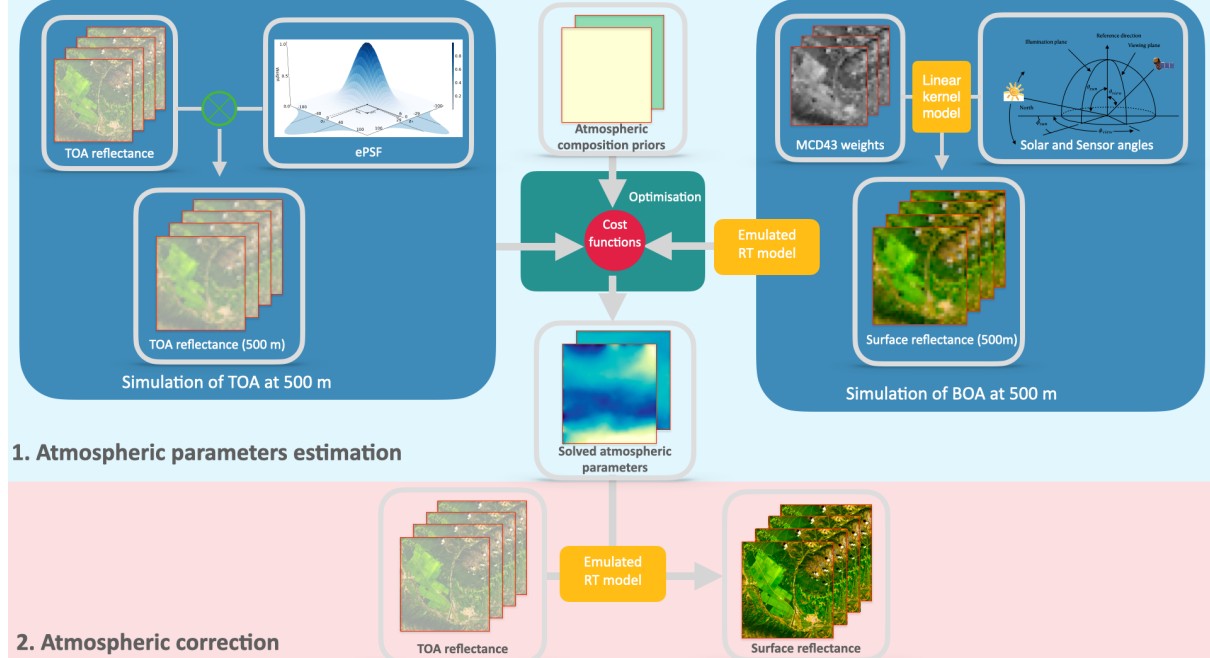

Figure 1: Schematic diagram of the SIAC processing chain.

We then use this to map from TOA to BOA reflectance (with associated uncertainty) at the native TOA data (S2/L8) resolution. The method has the following steps, also summarised as a flowchart in Fig. 1. For the target observational dataset (S2/L8 here) with given imaging location, geometry and spectral bands, we calculate:

1. Atmospheric parameters estimation

    (a) Sect. 2.3.1: simulation of TOA reflectance $Y_c$ at 500 m resolution from observations $Y$, scaled with a calibrated MODIS effective point spread function (ePSF) model.

    (b) Sect. 2.3.2: simulation of BOA reflectance $R_b$ at 500 m from MODIS MCD43A1 product, mapped to target sensor spectral bands for sample pixels.

    (c) Sect. 2.4: development of atmospheric composition priors, estimates of AOT and TCWV in $X_b$ from CAMS data, with a spatial correlation model on $X_b$.

    (d) Sect. 2.5: MAP estimate of the atmospheric parameters $X$ given $Y_c$, $R_b$, $X_b$.

2. Atmospheric correction

    (a) Sect. 2.6: application of $X$ to correct observed TOA reflectance $Y$ to *a posteriori* estimate of BOA BRF $R$.





| MODIS | | S2 | | L8 | |
|---|---|---|---|---|---|
| Band No. | Wavelength (nm) | Band No. | Wavelength (nm) | Band No. | Wavelength (nm) |
| 3 | 459-479 | 2 | 457-523 | 2 | 452-512 |
| 4 | 545-565 | 3 | 542-578 | 3 | 533-590 |
| 1 | 620-670 | 4 | 650-680 | 4 | 636-673 |
| 2 | 841-876 | 8A | 855-875 | 5 | 851-879 |
| | | 9 | 931-958 | | |
| 6 | 1628-1652 | 11 | 1565-1655 | 6 | 1566-1651 |
| 7 | 2105-2155 | 12 | 2100-2280 | 7 | 2107-2294 |

Table 3: MODIS, S2 and L8 bands used in SIAC for the atmospheric parameters retrieval.

## 2.3 Observational Constraint

We can express the observational negative log likelihood:

$$\mathcal{J}_{obs} = \frac{1}{2}(\hat{y} - y_c)^\top C_{\hat{y}}^{-1}(\hat{y} - y_c) \tag{2}$$

Here, $^\top$ is the matrix transpose operator and $^{-1}$ the matrix inverse operator. Calculation of $\mathcal{J}_{obs}$ as a function of variables in $X$ requires confronting a set of observations $y_c$ with modelled estimates $\hat{y}$ relative to uncertainty in these, expressed as $C_{\hat{y}}$ here. We derive these terms below.

### 100 2.3.1 TOA observations

The main data controlling the estimation of $X$ (and so $R$) are medium resolution observations $Y$ of TOA spectral reflectance from S2/L8, on a level 1C grid $G_m$ used to form the observational constraint above over wavebands $\Lambda_m$ (Table 3). There is a mostly close match between the medium resolution wavebands and those of the MODIS data used for constraints for S2/L8 so this is mostly a small correction, but we do not require this to be an exact overlap (hence the 'sensor invariant' nature of

SIAC) as a spectral mapping of information is applied (see Supplementary D). For S2, the water absorption band B09 provides information on TCWV, which is enabled by providing an estimate of BOA reflectance from MODIS by this mapping, even though B09 is not close to MODIS bands. Uncertainty from the spectral mapping is explicitly treated in the SIAC framework.

We need TOA observational constraints $Y_c$ to drive Eq. (2). The atmospheric state $X$ is defined at coarse resolution over grid $G_c$, so the observational likelihood term must also be defined at the same scale. This involves mapping valid observations

from $Y$ on grid $G_m$, to coarse resolution equivalents $Y_c$ on grid $G_c$. This is achieved using the effective Point Spread Function (ePSF) of the MODIS product following the approach of Mira et al. (2015), as described in Supplementary E. We ignore uncertainty associated with this aggregated reflectance in the estimation of $X$ via Eq. (2), assuming it is small compared to the atmospheric model uncertainty (below).



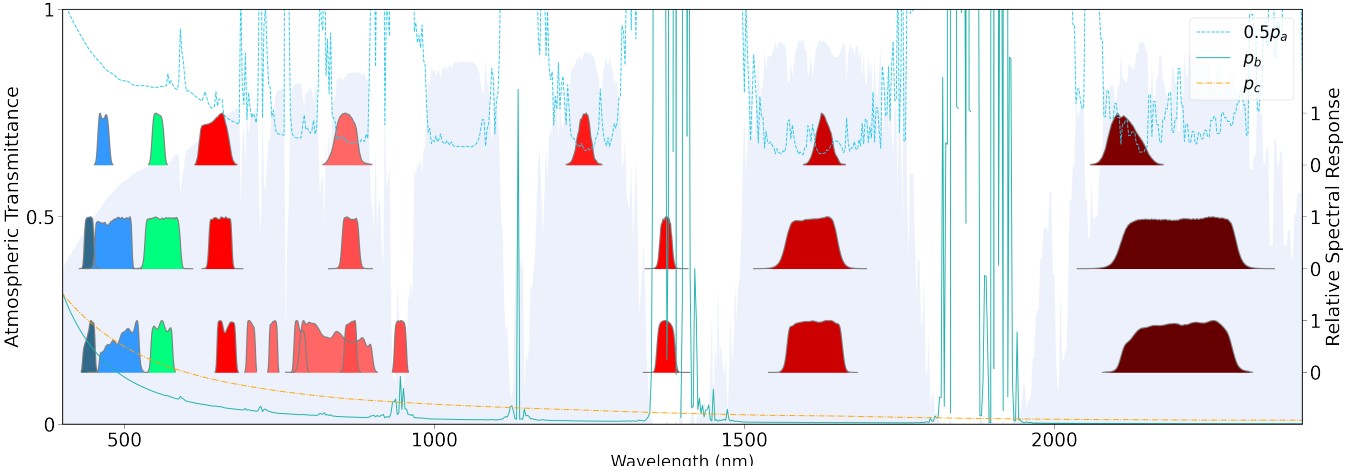

Figure 2: From the top to bottom are the MODIS, L8 and S2 relative spectral response function for each band, and the background is the atmospheric transmittance processed by 6S with US62 atmosphere profile and continental aerosol model with a $AOT$ value of 0.2 at $550\,\text{nm}$.

### 2.3.2 Modelling TOA reflectance

We need an estimate of TOA reflectance $\hat{Y}$ given atmospheric state $X$ to calculate $\mathcal{J}_{obs}$ in Eq. (2) which is provided by a radiative transfer (RT) model. In this paper, we follow other current approaches to this for medium resolution data by assuming the surface is Lambertian and that each pixel can be treated as independent. Under these assumptions, $\hat{y}$ is expressed by the 'simple-form' relationship described for the 6S RT model by (Vermote et al., 1997b):

$$\hat{y} = H(X_{a_c}) = r^{\uparrow} + \overline{\overline{t}}^{\downarrow}\overline{\overline{t}}^{\uparrow}\frac{r_b}{1 - \overline{\overline{r}}^{\downarrow}r_b} = \frac{r_b(p_b p_c - 1) - p_b}{r_b p_a p_c - p_a} \tag{3}$$

where: $p_a = 1/\overline{\overline{t}}^{\downarrow}\overline{\overline{t}}^{\uparrow}, p_b = r^{\uparrow}/\overline{\overline{t}}^{\downarrow}\overline{\overline{t}}^{\uparrow}$ and $p_c = \overline{\overline{r}}^{\downarrow}$ and $X_{a_c}$ is an augmented state vector defined in Table 2, on grid $G_c$. The terms $p_i, i \in a, b, c$ are lumped parameters for each waveband derived from 6Sv (Vermote and Kotchenova, 2008). Clearly, $p_a \geq 1$ and depends on pathlengths in the atmosphere. Outside of the strong absorption bands (L8 band 6, S2 bands B09, B11) it has a general pattern of decreasing with increasing wavelength as illustrated in Fig. 2. The path reflectance normalised by transmittance, $p_b$ and spherical albedo $p_c$ show a similar spectral trend and become small outside of visible wavelengths. $p_c$

impacts multiple scattering between land surface and aerosol layer in the atmosphere and is manifested as a slight curve in the relationship between $r_b$ and $y$. Under the Lambertian assumption currently implemented in SIAC, these terms fully define the mapping from BOA BRF $r_b$ to TOA BRF $\hat{y}$ as well as the inverse (estimating $r$ from $y$). Within SIAC they are calculated over a wide range of conditions using 6SV2.1 and emulated following Gómez-Dans et al. (2016) using fully connected Artificial Neural Networks (ANNs) for efficiency (Specht et al., 1991). This provides a fast surrogate model of the $p_i$ terms as a function

of variables in $X_{a_c}$. The derivatives of $p_i$ with respect to state variables in $X$ are calculated with back propagation of the ANN



(Hecht-Nielsen, 1992). These are used to estimate derivatives of $H(X_{a_c})$ for cost function minimisation and propagation of uncertainties. In the current version of SIAC, we assume that atmospheric profiles used in 6S are from the US62 the aerosol type is 'continental' Vermote et al. (1997b). This may cause errors when conditions strongly depart from this (e.g. urban, maritime or biomass burning conditions (Tirelli et al., 2015)).

Direct calculation of TOA reflectance $\hat{y}$ needs an estimate of $r_b$ for comparisons with observations $y$ in Eq. (2). Many algorithms take a spectral approach to the problem, assuming that the ratio in TOA reflectance between visible and SWIR bands over dark dense vegetation (DDV) targets is constant (Vermote and Saleous, 2006; Kaufman et al., 1997; Vermote et al., 1997a; Remer et al., 2005; Levy et al., 2007b, a). Most operational algorithms for S2 and L8 are based on this, including LEDAPS for Landsat 4-7 (Masek et al., 2012), LaSRC for Landsat 8 (L8) (Vermote et al., 2016), Sen2Cor for S2 (Louis
et al., 2016), MAJA for S2 (Hagolle et al., 2015a), etc. This constraint can however be of limited value if suitable DDV targets cannot be found well-dispersed in the scene. Other relevant approaches applied to coarse resolution data find alternative methods to estimate surface reflectance: the Deep Blue (DB) method (Hsu et al., 2013) uses a coarse resolution seasonal global reflectance database for blue and red wavelengths over bright surfaces to extend the range of conditions that can be used; MAIAC (Lyapustin et al., 2018) develops an expectation from a time series of observations; and Guanter et al. (2007) uses a
set of spectral basis functions that need to be solved for for each observational constraint sample. A variation of that is the use of a wider set of spectral basis functions used in processing hyperspectral data by Hou et al. (2020).

    In SIAC, we avoid the sampling limitations of DDV, and take advantage of these other ideas of providing a dynamic and globally-applicable expectation of surface reflectance. We use the MODIS MCD43A1 BRDF/albedo (collection 6) product (Schaaf et al., 2002; Schaaf and Wang, 2015) to achieve this and derive an *a priori* model of surface reflectance for all target
wavebands for the viewing and illumination angles $\Omega_v$, $\Omega_s$ respectively of $Y_c$. For relative day index $d$, this gives $r_b(d)$ at MODIS wavebands $\Lambda_c$, via the Ross-Thick/Li-Sparse-Reciprocal (RTLSR) linear kernel models (Wanner et al., 1997) and the values of the model parameters for relative day $d$:

$$r_b(d, \Lambda_c) = f_{iso}(d) + f_{vol}(d)k_{vol}(\Omega_v, \Omega_s) + f_{geo}(d)k_{geo}(\Omega_v, \Omega_s) \tag{4}$$

    Samples of this around the day of the observation $d$ are used to provide a gap-filled uncertainty-quantified estimate of
$r_b(\Lambda_c)$ detailed in Appendix A. This is mapped to the target (S2/MSI or L8/OLI) waveband set $\Lambda_m$ as $r_b(\Lambda_m)$ as given in Supplementary D. The framework can tolerate incomplete coverage of $\hat{Y}$, so we filter for plausible constraints from the MODIS data as described in Appendix C to avoid gross errors from inappropriate values of interpolated MODIS surface reflectance.

### 2.4   *A priori* constraint on atmsopheric state

We can express the *a priori* negative log likelihood:

$$\mathcal{J}_{prior} = \frac{1}{2}(x - x_b)^T C_{x_b}^{-1}(x - x_b) \tag{5}$$



This gives a constraint based on a background (*a priori*) estimate of atmospheric state, $X_b$. In SIAC we use information from European Centre for Medium-Range Weather Forecasts (ECMWF) CAMS Near-real-time services (Morcrette et al., 2009; Benedetti et al., 2009) for estimates of atmospheric composition parameters $AOT$ at $550\,\text{nm}$, total column water vapour

and total column of Ozone in $X_b$ and $X_{a_c}$. These are at a coarse spatial resolution on a 40 km grid, but we need $X_b$ on a 500 m grid to match with $r_b$, so the data are interpolated to 500 m resolution. We constrain with a sub-40 km spatial correlation structure using a Markov process covariance after Rodgers (2000). This has two free parameters, the variance $\sigma^2_{x_b}$ and relative length scale. Since the covariance appears in the constraint Eq. (5) in inverse form, we use a fast approximation this is derived from Rodgers (2000) and implemented as a first-order spatial difference constraint defined in matrix $D$.

$$C^{-1}_{x_b} = \frac{1}{\sigma^2_{x_b} k^2} \left( I + \gamma^2 D^T D \right) \tag{6}$$

Here, $I$ is the identity matrix, $k$ a normalising scale factor given in Eq. (B3), and $\gamma$ an implicit function of the relative length scale that controls the degree of spatial smoothness. Numerical values used for uncertainty in SIAC are given in Appendix B.

### 2.5 MAP estimate of $X$

We obtain the MAP estimate of $x$ by minimising $\mathcal{J}$ in Eq. (1) with respect to $X$. This is done simultaneously for all samples in

the grid $G_c$ of $X$ using the efficient L-BFGS-B gradient descent algorithm (Byrd et al., 1995; Zhu et al., 1997). The approach and details follow Lewis et al. (2012a), using the derivatives of $\mathcal{J}$ with respect to $X$. Multi-grid methods (following e.g. Briggs et al. (2000)) are used to iteratively provide spatially-refined solutions. This greatly improves convergence rates in the optimisation over the large dimensional state vector of $X$. The uncertainty in $X$, $C_x$ is calculated as in Appendix B3.

### 2.6 Atmospheric correction

The mapping from $Y$ to $R$ given $X$ at medium (10-60 m) spatial resolution on the grid $G_m$ is achieved rearranging the terms in Eq. (3) to give $r$ (Vermote et al., 2006):

$$r = \frac{p_a y - p_b}{1 + p_c \left( p_a y - p_b \right)} \tag{7}$$

We assume that mean atmospheric parameters $x$ and the auxiliary data (Ozone and elevation) are constant at resolutions higher that the MODIS spatial grid $G_c$, so the emulated $p_{a,b,c}$ can be applied to all S2/L8 pixels within each grid cell of $G_c$

to derive the mean surface reflectance $r$. The TOA uncertainty is taken to be 5% (Barsi et al., 2018b; Lamquin et al., 2019; MPC Team, 2021) for both S2 and L8 and independent for each waveband. This is the threshold uncertainty value for S2 TOA reflectance. The calculation of per pixel uncertainty in $r$ uses partial derivatives of $r$ with respect to atmospheric parameters $x$ and TOA reflectance $y$ (Ku, 1966), as shown in Appendix B4. The per-pixel reflectance uncertainty derived in this way and propagated from uncertainty in the atmospheric parameters and the measurements, is an important feature of SIAC.





## 3 Materials and Method

### 3.1 Study region and Datasets

| Dataset | Usage | Reference | Notes |
|---------|-------|-----------|-------|
| S2 | TOA reflectance | ESA (2015) | |
| L8 | TOA reflectance | Roy et al. (2014) | |
| ASTER Global DEM | Per pixel elevation | Tachikawa et al. (2011) | Horizontal resolution of 75 meters covering 83° north (N) and 83° south (S) latitudes |
| ESA global water mask | Water mask | (ESA, 2017) | |
| MCD43A1 | Surface reflectance expectation | Schaaf et al. (2002); Schaaf and Wang (2015) | |
| CAMS | Prior for AOT and TCWV | Morcrette et al. (2009); Benedetti et al. (2009) | |
| Spectral libraries | Spectral mapping from MODIS to target sensor | Pearson et al. (2017), Baldridge et al. (2009), Ilehag et al. (2019), Garrity and Bindraban (2004) | USGS V7, ASTER, KLUM, ICRAF-ISRIC |
| AERONET | Validation of retrieved AOT and TCWV | Giles et al. (2019); AERONET (2021) | Data from 2017-19 |
| RadCalNet | Validation of surface reflectance | Bouvet et al. (2019) | Data from 2017-19 |

Table 4: Datasets used in SIAC.

We validate using SIAC-derived atmospheric composition to estimate surface reflectance over globally representative sites for the years 2017-2019. We use S2 and L8 granules over the more than 400 AERONET sites in Fig. 3, as well as granules encompassing three RadCalNet sites (Railroad Valley Playa, La Crau and Gobabeb sites). This gives more than 3000 S2 tiles and more than 2500 L8 tiles in the evaluation. The datasets used in this study, with comments on their use are listed in Table 4.

### 3.2 AERONET

The AERONET (AErosol RObotic NETwork) (Giles et al., 2019; AERONET, 2021) program is a federation of ground-based remote sensing aerosol networks and provides globally distributed observations of $AOT$, inversion products, and precipitable water. It has long been used as ground truth aerosol measurements and used for the validation of various satellite inversions aerosol products. Atmospheric measurements from AERONET instruments were interpolated in time to get a estimates corresponding to each satellite overpass. AOT at $550\,\mathrm{nm}$ was estimated using AERONET spectral log-transformed data with a second order polynomial between $400\,\mathrm{nm}$ and $860\,\mathrm{nm}$ following Kaufman (1993); Li et al. (2012). The measurement uncertainty of AOT from AERONET is taken to be 0.01 (Eck et al., 1999; Sayer et al., 2020), and that for TCWV 0.15% (Pérez-Ramírez et al., 2014).





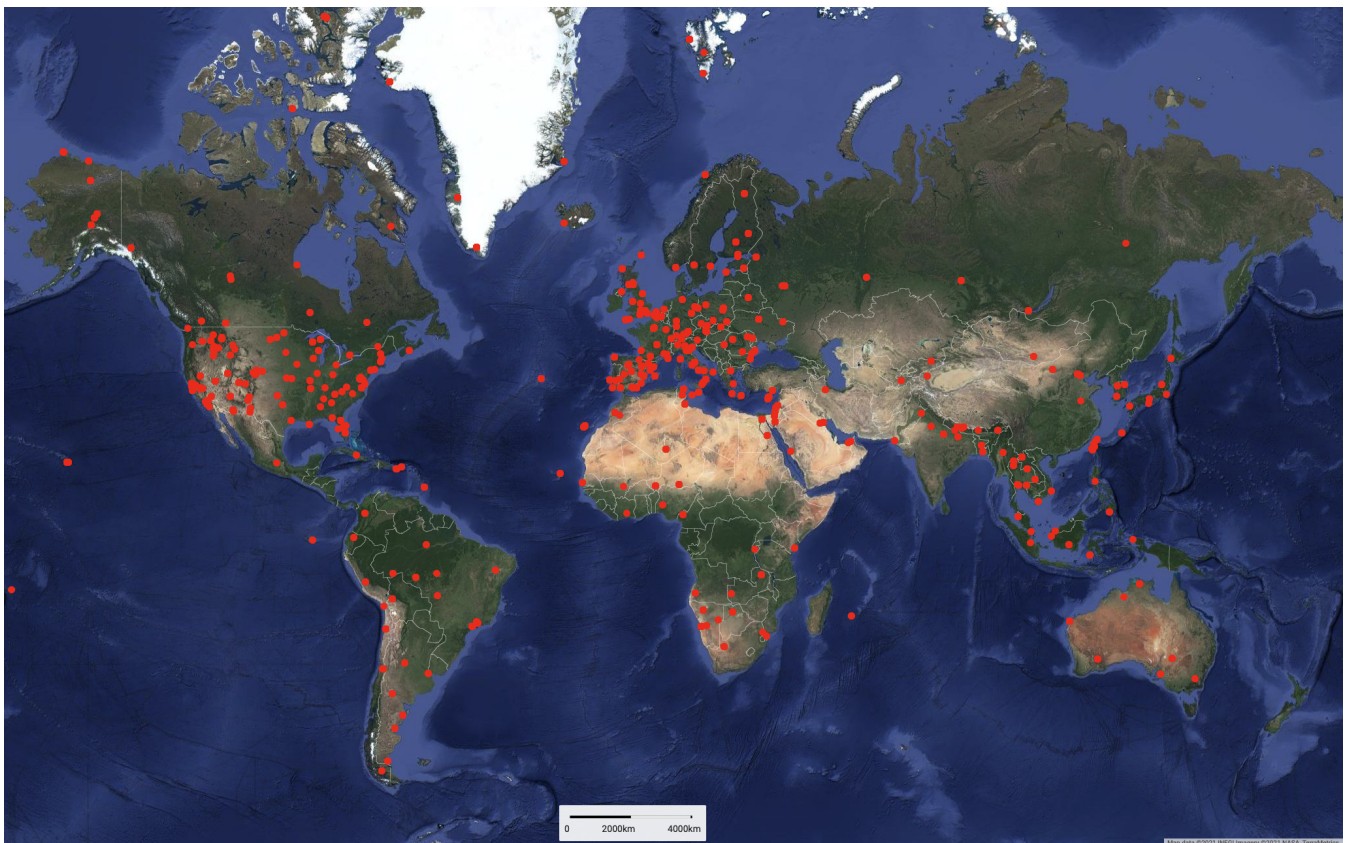

Figure 3: Globally distributed AERONET sites (red dots) used in this study for the validation of retrieved atmospheric parameters.

## 3.3 RadCalNet data

The Working Group on Calibration and Validation (WGCV) of the Committee on Earth Observation Satellites (CEOS) has started providing ground surface reflectance data through the Radiometric Calibration Network portal RadCalNet (Bouvet et al., 2019) since 2018, with measurements taken earlier 2013 in the US Railroad Playa Valley site. RadCalNet provides nadir-view, top-of-atmosphere reflectance at 30 minute intervals from 9am to 3pm local standard time at $10\,\mathrm{nm}$ intervals from $400\,\mathrm{nm}$ to $2500\,\mathrm{nm}$, which is calculated from ground nadir-view reflectance measurements, and atmospheric measurements such as surface pressure, columnar water vapour, columnar ozone, aerosol optical depth and the Angstrom coefficient. TOA reflectance is simulated by propagating the measured surface reflectance through the atmosphere using the MODTRAN radiative transfer model, parameterised by measured local atmospheric composition measurements.

Here, we compare the SIAC-corrected data with measurements from three RadCalNet sites: the ESA/CNES site in Gobabeb (Namibia), the CNES site in La Crau (France) and the University of Arizona's site at Railroad Playa Valley (Nevada, United





States), as these three sites measure over the entire solar reflective spectrum. Railroad Playa Valley is a high-desert playa surrounded by mountains to the East and West, La Crau has a thin pebbly soil with sparse vegetation cover, and Gobabeb is over gravel plains. The Area of Interest (AOI) of the radiometric measurements for the sites is taken to be $30m \times 30m$ for Gobabeb and La Crau, but $1km \times 1km$ the Railroad Valley Playa. The appropriate (S2 or L8) sensor spectral response

functions are applied to the RadCalNet hyperspectral measurements that are closest in time to the S2 and L8 acquisitions to derive RadCalNet estimates of BOA reflectance in S2/L8 wavebands. Gross mis-matches due to cloud or other artifacts are filtered by comparing RadCalNet TOA reflectance (provided by RadCalNet) estimates with S2/L8 TOA reflectances. If the S2/L8 TOA data fall outside a tolerance of 10% of the RadCalNet TOA reflectances, we remove the sample from the comparison. This ends up with 273 S2 scenes and 72 L8 scenes over the RadCalNet sites, where 100 S2 scenes and 36 L8

scenes over Gobabeb, 93 S2 scenes and 19 L8 scenes over La Crau, and 80 S2 scenes and 17 L8 scenes over Railroad Valley Playa.

### 3.4  Sentinel 2 and Landsat 8

Sentinel 2A (S2A) and Sentinel 2B(S2B) were launched on 23/06/2015 and 07/03/2017 respectively. A single satellite revisits the equator every 10 days, while a constellation of two satellites achieves an equatorial revisit time of 5 days, decreasing

to 2-3 days at mid-latitudes. Each S2 has $10\,m$, $20\,m$ and $60\,m$ spatial resolution Multi-Spectral Instrument (MSI), with 13 spectral bands ranging from $443\,nm$ to $2190\,nm$. Identical to S2A and S2B, Sentinel 2C (S2C) is expected to be launched at the beginning of 2024 in which case S2A will be retired (ESA, 2021a).

The Landsat project has provided the longest temporal record of moderate resolution multi-spectral data over the Earth surface. Landsat 8 was launched at 11/02/2013, having a global revisit time of 16 days with 8 days offset to Landsat 7 for 8-day

repeated coverage. Two push-broom sensors: the Operational Land Imager (OLI) and the Thermal Infrared Sensor (TIRS) are mounted on the platform to provide multi-spectral and thermal observations of the earth surface at $30\,m$ and $100\,m$ resolution respectively. OLI has 9 spectral bands, among which band 8 is panchromatic and has a spatial resolution of $15\,m$. At the time of writing, Landsat 9 (Masek et al., 2020) had recently been launched (27/09/2021), but operational data is just coming online (USGS, 2021).

Both products provide projected and calibrated TOA reflectance datasets. Sentinel 2 products were obtained from the Copernicus Open Access Hub, and the L8 products from the USGS EarthExplorer. The spectral characteristics of S2/MSI and L8/OLI, along with the MODIS land wavebands used in SIAC are shown in Fig. 2 and Table 3.

We process all near simultaneous (maximum 1 hour apart) scenes/tiles from S2 and L8 over the years 2017 to 2019 over the AERONET sites illustrated in Fig. 3. This gives 2635 S2 and 1922 L8 scenes and 3472 point samples.

### 3.5  Validation approach and metrics

We want to evaluate how well SIAC estimates mean BOA BRF and associated uncertainty over S2/L8 wavebands. We can validate mean reflectance against measurements for some conditions using RadCalNet data. But we can also gain confidence in the results by validating interim products (atmospheric parameters), testing uncertainty via the discrepancy principle (Sayer



et al., 2020), and examining patterns in uncertainty behaviour. Since we estimate surface reflectance from both S2 and L8

sensors, and since these have some very similar wavebands, it is also worthwhile to look at the consistency of results between

the sensors for samples over the same conditions.

We define residuals between values estimated from SIAC and measurements:

$$\Delta_{x_{atmo}} = x_{atm} - x_{aeronet} \tag{8}$$

$$\Delta_{RadCalNet} = r - r_{RadCalNet} \tag{9}$$

for the residual $\Delta_{x_{atmo}}$ for atmospheric parameters calculated from SIAC ($x_{atm}$) and AERONET ($x_{aeronet}$) and $\Delta_{RadCalNet}$

for that between SIAC estimated reflectance and RadCalNet measurements $r_{RadCalNet}$.

We define standardised residuals:

$$\epsilon_{x_{atm}} = \frac{\Delta_{x_{atmo}}}{\sqrt{\sigma_{x_{atm}}^2 + \sigma_{x_{aero}}^2}} \tag{10}$$

$$\epsilon_r = \frac{\Delta_{RadCalNet}}{\sqrt{\sigma_r^2 + \sigma_{r_{RadCalNet}}^2}} \tag{11}$$

where $\sigma_{x_{atm}}$ and $\sigma_{x_{aero}}$ are the uncertainty in the SIAC retrievals of atmospheric parameters and aeronet measurements

respectively. $\sigma_r$ and $\sigma_{r_{RadCalNet}}$ are the uncertainty in the SIAC retrievals of surface reflectance and that of the RadCalNet

measurements respectively. Assuming Gaussian distributions, we would expect the mean of $\epsilon_{x_{atm}}$ or $\epsilon_r$ to be zero and the

standard deviation 1 over a large number of samples. We follow Doxani et al. (2018) in calculating accuracy (A) and uncertainty

(U) metrics against AERONET and RadCalNet observations through:

$$A = \frac{1}{n} \sum_{i=1}^{i=n} \Delta \tag{12}$$

$$U^2 = \frac{1}{n} \sum_{i=1}^{i=n} \Delta^2 \tag{13}$$

where $n$ is the total number of samples in a comparison and $\Delta$ is $\Delta_{x_{atmo}}$ or $\Delta_{RadCalNet}$ as appropriate. The related measure,

precision (P), is given by $P^2 = \left(\frac{n-1}{n}\right) U^2 - A^2$. We recognise A as a measure of bias, and U as the root mean squared error

(RMSE). We assess SIAC against the threshold requirements in Table 1. For SIAC results be *within specification*, we would

expect 68% to fall at or below the threshold value (assuming a Gaussian distribution) where samples or distributions are

concerned. Where we calculate U, we would expect it to lie on or below the threshold value.





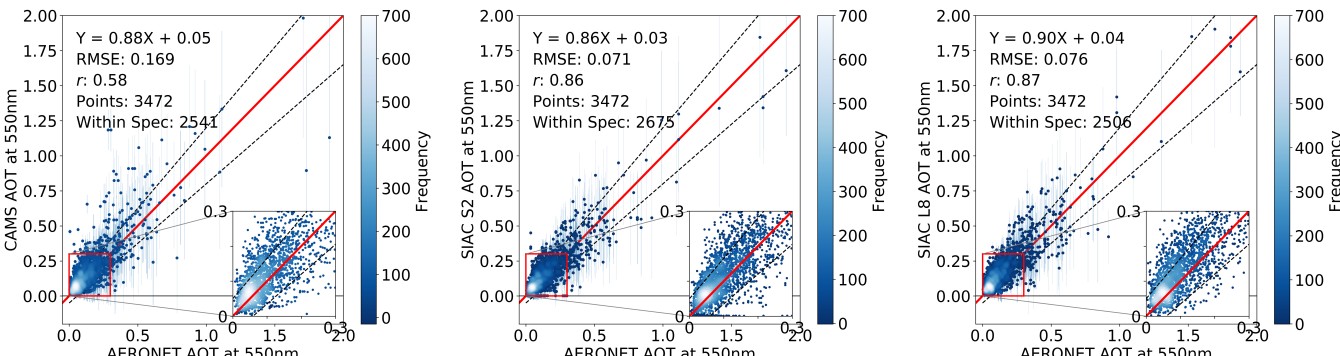

Figure 4: AOT validation against AERONET measurements from CAMS (left), SIAC S2 (middle) and SIAC L8 (right) over 3466 matches, where the vertical lines of each point is the uncertainty of solved AOT values and the horizontal error bars are AERONET measurement uncertainty of 0.01. On three panels, the inset plot shows the region region marked by the red square in more detail, with $0 \leq AOT \leq 0.3$. The threshold uncertainty is shown as black dashed lines in the figure.

# 4 Results

## 4.1 Validation of mean atmospheric composition over AERONET

We compare the 3472 S2/L8 samples over AERONET sites with independent *in situ* measurements of AOT and TCWV.
Examples of the retrieved scene atmospheric parameters are given in Fig. F1 - Fig. F3 in Supplementary F. Since we use an *a priori* constraint on atmospheric state $X_b$ in estimating $X$, we also assess $X_b$ against the AERONET measurements to see what improvement the medium resolution observations offers in this context. Comparisons of CAMS and SIAC AOT with AERONET measurements are shown in Fig. 4, with more detail for A, P and U (along with the number of samples used in each bin) given in Fig. 5. The threshold uncertainty (Table 1) is shown on the plots.

Over all AOT values (Fig. 4) for the *a priori* CAMS data, more than 73% of samples are already within the threshold specification, which is slightly better than the 68% expected. This increases to 77% for S2 processing, but is slightly reduced, to 72% for L8. The correlation coefficient is reasonably high for CAMS (0.58) but dramatically improved by the data assimilation, to 0.86 or better for S2 and L8. The regression for all cases is similar, with a slope is slightly below unity (0.86-0.90) and a small intercept (0.03-0.05). The root mean squared error (RMSE, equivalent to the metric U over all samples) is moderately
large, at 0.169 for CAMS, but reduced to 0.071-0.076 by the assimilation. The A, P and U plot shows that bias (A) is low and uncertainty and precision are close to the expected error for low values of AOT for both sensors, with S2 mostly slightly better than L8. The results are more variable and sometimes out of specification for higher values of AOT, but the sample size is small for those cases.

    Comparisons of CAMS and SIAC TCWV with AERONET measurements are shown in Figs. 6 and 7. Over all values of
TCVW, 86% of the CAMS data are within specification. This is essentially the same after assimilation of L8 data, but increases to 91% for S2. The regressions for CAMS and L8 TCWV against AERONET have a slope close to unity and a low magnitude




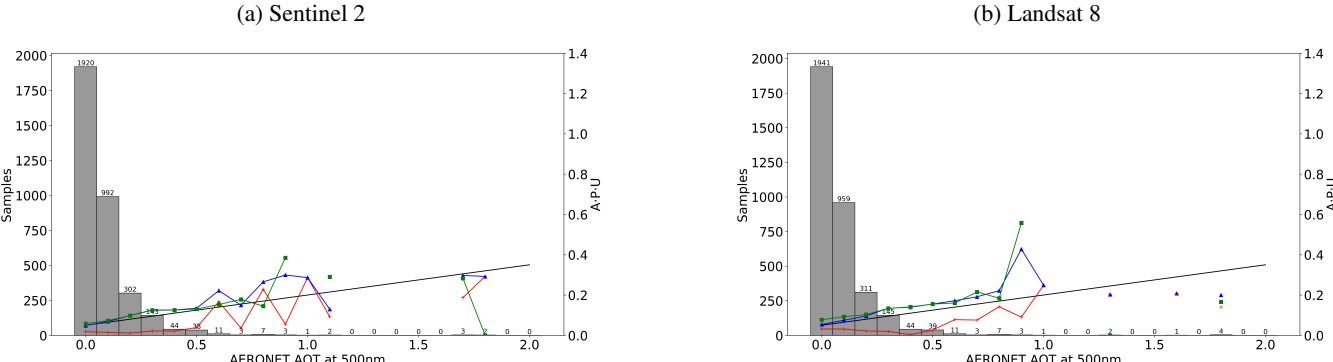

Figure 5: The accuracy (A) (red), precision (P) (green) and uncertainty (U) (blue) validation of AOT against AERONET measurements. Threshold uncertainty is shown as black lines in the figure.

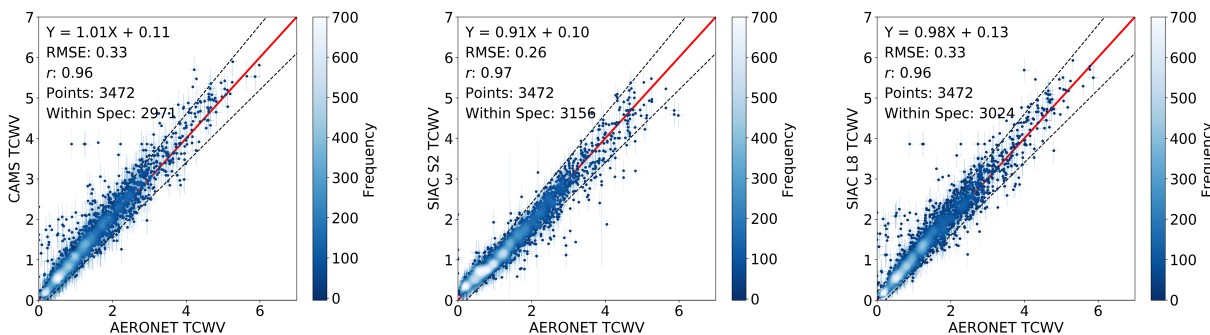

Figure 6: TCWV ($g/cm^2$) validation against AERONET measurements from CAMS (left), SIAC S2 (middle) and SIAC L8 (right) over 3466 matches, where the vertical lines of each point is the uncertainty of solved TCWV values and the horizontal error bars are 15% of AERONET TCWV values. Threshold uncertainty is shown as dashed lines in the figure.

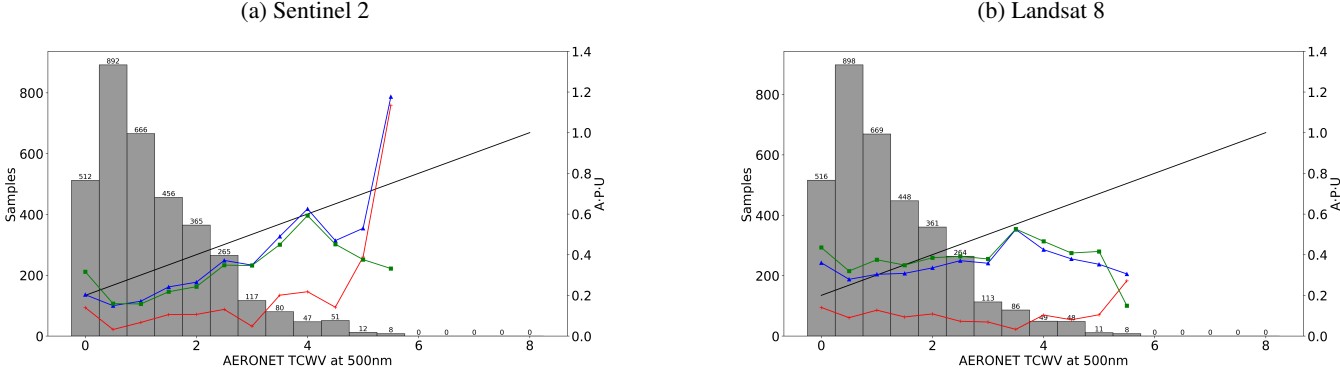

Figure 7: The accuracy (A) (red), precision (P) (green) and uncertainty (U) (blue) validation of TCWV against AERONET measurements. Threshold uncertainty is shown as black lines in the figure.





of intercept. The slope of the relationship is slightly poorer for S2, and 0.91. The A, P and U plot shows S2 results are within specification for the vast majority of values of TCWV, with poorer A and U only for the highest value and P slightly out of specification for the lowest value. For L8, the results are very similar to those from CAMS alone. In this case, all values

are within expected error, other than the precision and uncertainty for low TCWV. In summary, the CAMS and L8 *a priori* data are very similar (very low impact of the observations), and the results good for all but the lowest values of TCWV. The assimilation of S2 data (mainly from band B09) improves these lower values, but seems to cause poorer result for the highest value of TCWV though this may be because of the small sample number.

## 4.2 Consistency check for S2 and L8

The S2 and L8 scenes over AERONET sites described above were atmospherically corrected to $R$ using SIAC. Since the overpass times between sensors are within one hour, we expect the surface reflectance in overlapping spectral regions from both sensors to be highly correlated and can use this to test consistency between sensors. Differences in spatial coverage, acquisition geometry, spectral sampling and other sensor characteristics may impact this, but we will assume them to be small.

Pixels within a $2400 \times 2400 m^2$ area around the AERONET sites in the S2 and L8 scenes presented above were considered

for comparison. To account for geolocation errors and differences in spatial resolution, the L8 data were reprojected to the S2 reference system and all data spatially averaged to $60\,\mathrm{m}$ resolution. A filtering for cloud, shadow and any large changes between scene acquisitions was then applied. Rather than relying on the cloud/shadow masks for this, we use compatibility in *TOA reflectance* to select candidate pixels. According to studies (Gascon et al., 2017; Barsi et al., 2018a; Helder et al., 2018; Pahlevan et al., 2019; Lamquin et al., 2019) and operational validation reports (Clerc et al., 2021), the agreement of nearby

spectral bands (Table 3) should be better than 5%. Since we allow a larger temporal gap between S2 and L8 than some of these studies (1 hour), we use a filter on a threshold of $10\% + 0.01$ between S2 and L8 TOA reflectance. This leaves around $3 \times 10^6$ pixels for comparison.

The results are shown in Fig. 8 as a set of two-dimensional histograms. The reflectances are highly correlated (coefficient of determination $r > 0.98$ for all bands, and $r > 0.99$ for bands in the NIR and SWIR regions), with a small RMSE ($RMSE <$

0.012). The bias is very small (less than 0.0016 for all bands), and the slope is between 0.96 (blue band) and 1 (NIR band). The error bars of S2 and L8 bands are slightly larger than the 10% + 0.01 used to filter the TOA reflectances. This slight increase in the difference between S2 and L8 surface reflectance comes from the increase in uncertainty during the atmospheric correction process, but this is any case small.

## 4.3 Validation of uncertainty in atmospheric parameters

We need to verify that uncertainty values $\sigma_x$ calculated by SIAC are useful in characterising actual uncertainty. We approach this using the 'discrepency analysis' method suggested by Sayer et al. (2020) to check if the error distributions in the AERONET comparisons described above follow an expected distribution. We calculate standardised residuals $\epsilon_{x_{atm}}$ following Eq. (10) for AOT and TCWV for each sample. We know that different configurations and data and algorithmic effects mean that some retrievals will be more accurate than others, so here, we test our ability to identify this, by weighting the departure of SIAC





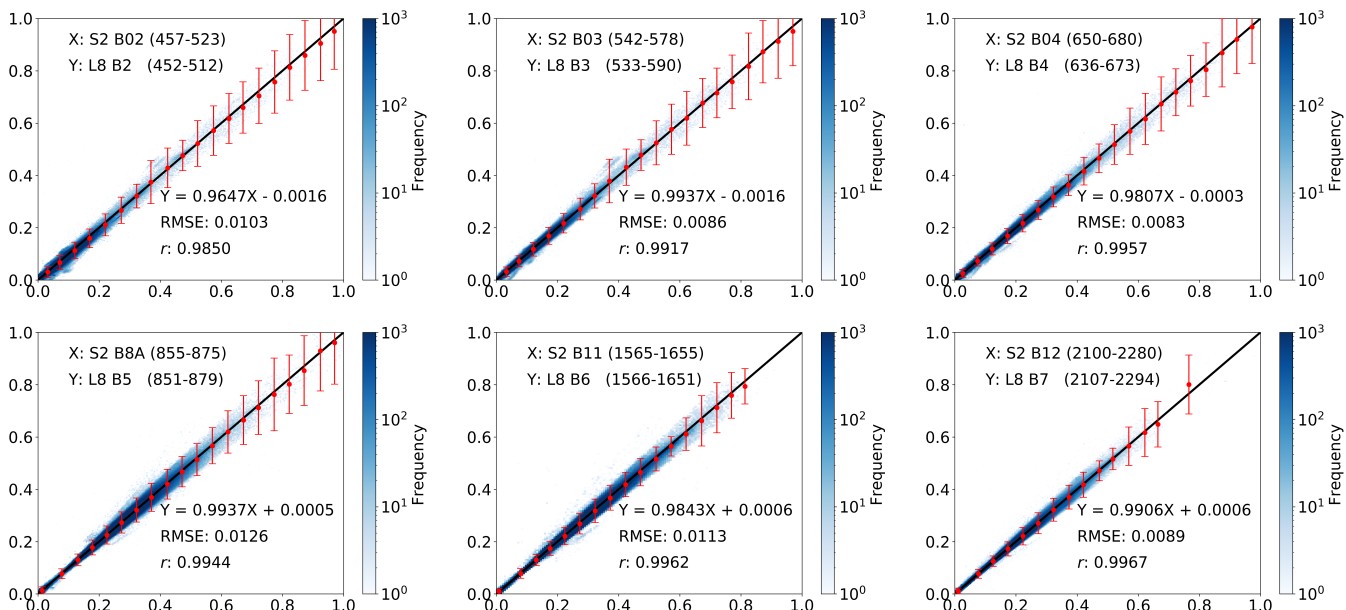

Figure 8: 2D histogram of surface reflectance after the atmospheric correction from 3466 S2 and L8 near simultaneous observation, and each subplots shows the results for the closest S2 and L8 bands. The colourbar is shown using a logarithmic scale. The red error bars are $3\sigma$ of the difference between the S2 and L8 corrected reflectance, which is computed at an interval of 0.05 from 0-1.

estimates from AERONET measurements. If we have grossly over-estimated uncertainty relative to actual discrepancy, then the standard deviation of the standardised residuals will be much less than one, and *vice-versa* if we under-estimate. A limitation of the assessment is that it can only calculated over the AERONET sites where an independent measurement is available. But still, if the results mainly follow the expected statistics, it shows that the magnitude of uncertainties calculated in SIAC are plausible.

The results for comparisons for S2 and L8 bands are shown in Sect. 4.3. Part of the *a priori* constraint in SIAC is the imposition of a degree of smoothness on the atmospheric parameters through the parameter $\gamma$ in the inverse covariance function in Eq. (6). Cross validation studies in Section G indicate that a wide range of $\gamma$ values may be appropriate here, but we use values of 5 for S2 and L8 AOT, 5 for S2 TCWV and 0.1 for L8 TCWV in SIAC. The scaling term $k$ in that Eq. (6) should mean that magnitude of uncertainty is not greatly affected by $\gamma$. But since many applications of this type of constraint (Dubovik et al., 2011; Lewis et al., 2012a; Govaerts and Luffarelli, 2018) don't explicitly apply such a normalisation, we test the impact of that here and examine the distribution of $\epsilon_{x_{atm}}$ over a range of $\gamma$ values in Fig 9(a-d).

The results show that for $\gamma < 10$, the standardised residual distribution of AOT for S2 and L8 remain broadly similar, i.e. the normalisation using $k$ seems to be effective. For S2, there is a small positive bias in AOT that decreases with increasing $\gamma$, but the standard deviation $\sigma$ is around 1.0. For L8 AOT, there is a small positive bias in all cases (larger than for S2) but $\sigma$ is





(a)                         (b)

S2 AOT normalised                    L8 AOT normalised

(c)                         (d)

S2 TCWV normalised                   L8 TCWV normalised

Figure 9: Standardised residual $\epsilon_{x_{atm}}$ distributions for S2 AOT (a), L8 AOT (b), S2 TCWV (c) and L8 TCWV (d) with $\gamma$ of [0.001, 0.1, 1, 5, 10, 100]. The red histogram distributions are the original normalised error distributions and the blue ones are the estimated Gaussian distribution from the distributions.



close to 1. Above $\gamma = 10$ for both, $\sigma$ increases, and becomes unrealistic for very high $\gamma$, so the normlisation is ineffective for extremely high values, possibly relating to boundary condition effects on Eq. (6) as an approximation to the intended inverse Markov process covariance function.

    The distributions of $\epsilon_{x_{atm}}$ for S2 TCWV retrieval show a slight overestimation in the TCWV uncertainty ($\sigma$ smaller than 1) but almost no bias for $\gamma < 10$. The L8 retrieval for TCWV is mainly controlled by the prior information and the normalised

error is close to 1, but with a broader distribution compared to S2 TCWV as no water absorption band available from L8 measurements. The behaviour of these distributions for TCWV again broadly confirms our choice of $\gamma$ for S2, but is seems that a higher value for L8 might be tolerated, and a compromise value of 5 might reasonably be used for $\gamma$ for all terms.

### 4.4   Validation of mean surface reflectance

  We validate mean SIAC reflectance by comparison with ground measurements over RadCalNet sites. We compare mean S2/L8

BOA reflectance from SIAC averaged over defined RadCalNet AOI boundaries with the RadCalNet estimates of BOA reflectance in Figs. 10, 11 and 12. Since TOA reflectance estimates are provided for the RadCalNet sites using observed surface reflectance and atmospheric parameters calculated with the 6S model, we also compare measured TOA reflectance for S2 and L8 with these. This provides context to interpret the both spectral signatures and any biases or other issues in the data. If there are mis-matches between the TOA datasets (e.g. from sensor calibration), since one is essentially a direct (S2/L8) measurement

and the other developed only with measurements from the RadCalNet sites, we would not expect to do better than that using SIAC where we have to estimate the atmospheric parameters.

    The agreement between the SIAC-retrieved surface reflectance and the reference measurements is very strong for all sites, with RMSEs values for the BOA products of around 3-5% of RadCalNet ground measurement reflectance over all wavebands. The correlation coefficient $r$ is very high ($> 0.94$) for all cases, and is seen to increase slightly between TOA and BOA

reflectance. The proportion of samples within the specification for TOA reflectance and SIAC corrected surface reflectance are very similar. For BOA there are 98% for S2 and 95% respectively within the specification for L8 over Gobabeb site, 88% for S2 and 87% for L8 over La Crau site, and 77% for S2 and 86% for L8 over Railroad Playa Valley site, so the results overall are well within the specification. Most samples outside of this can be attributed to the TOA reflectance being outside the RadCalNet TOA expectation limits. The patterns in the scatterplot of the small apparent biases in BOA reflectance are

mirrored in the TOA analysis, suggesting that these arise from factors extraneous to the atmospheric correction. Interestingly, the results obtained using only the *a priori* CAMS data (included in Supplementary I) show almost the same performance comparing against RadCalNet as mean SIAC reflectance retrievals.

    The best performance is found over Gobabeb, but the results are only slightly poorer for La Crau which has more variation in the pattern of spectral reflectance. The broader spread of results for the BOA analysis for Railroad Playa Valley is mimicked

in the TOA data. A per-band analysis of the ratio of SIAC BOA reflectance to measured ground data over each RadCalNet site is given in Fig. 13a and 13b. For Gobabeb, the SIAC surface reflectance is within 5% of RadCalNet ground measurements most of the time for all S2 and L8 bands, excluding the water absorption and deep blue bands (not shown). A similar situation



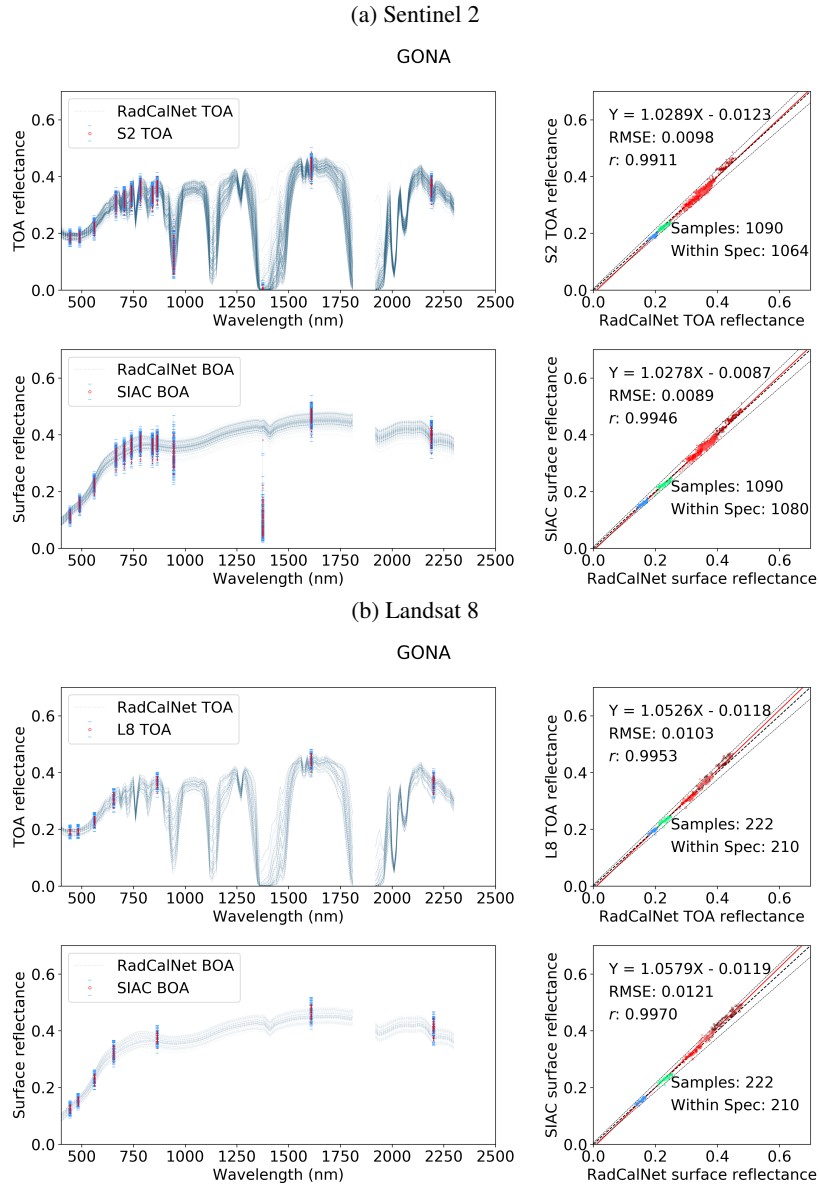

Figure 10: Comparison between the S2 (a) and L8 (b) TOA reflectance and RadCalNet simulated nadir-view TOA reflectance (top row), and the surface reflectance after correction against RadCalNet nadir-view surface reflectance (bottom row) at Gobabeb. The blue lines at left are different spectra measurement from RadCalNet and the red dot with blue error bars (1 standard deviation) are the TOA or surface and TOA reflectance with uncertainty. The EE is defined as $\pm(0.05TOA(BOA) + 0.005)$, denoted as black dash lines in the scatter plots. The regression line is draw as red line and the 1 to 1 reference line is draw as thick black dash line in the middle.



(a) Sentinel-2

LCFR



(b) Landsat 8

LCFR

Figure 11: Same as Fig. 10 but for La Crau site.



(a) Sentinel 2

RVUS



(b) Landsat 8

RVUS

Figure 12: Same as Fig. 10 but for Railroad Valley Playa site.





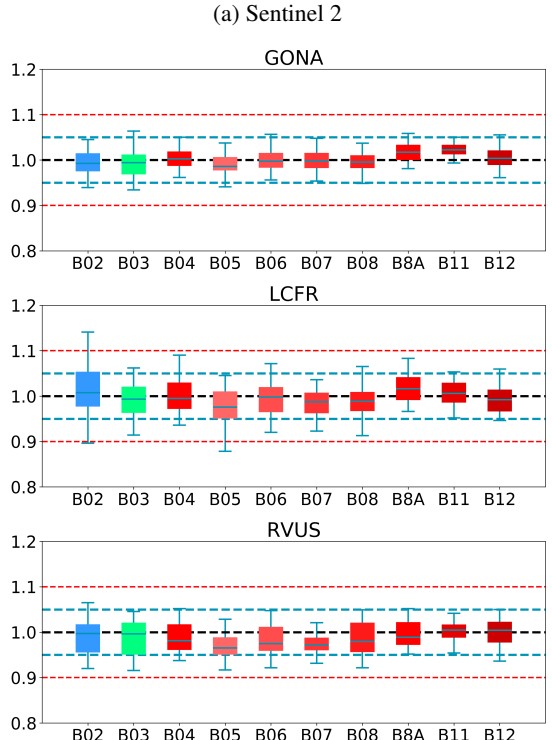

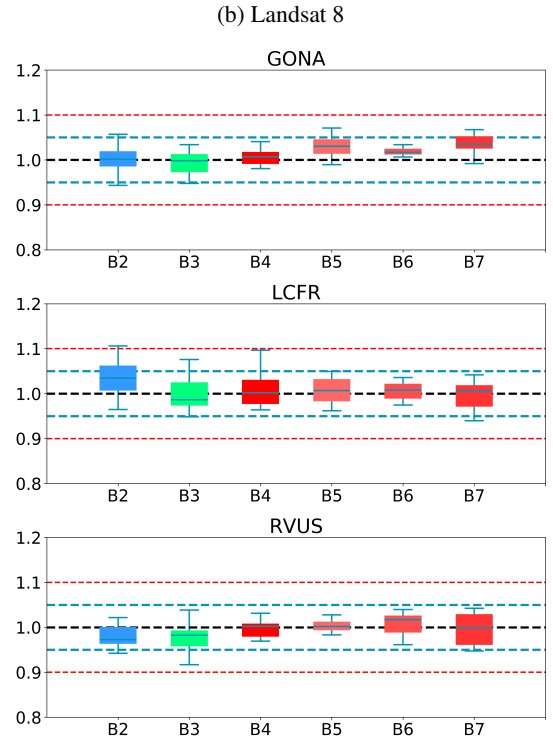

Figure 13: The ratio of SIAC corrected S2 (a) /L8 (b) surface reflectance to RadCalNet surface reflectance over three sites. The back dashed line in the middle is reference line of ratio 1, indicating the same values of SIAC corrected S2 (a) /L8 (b) surface reflectance and RadCalNet surface reflectance. Values above the reference line means positive bias (SIAC overestimates compared to RadCalNet), and below it means negative bias. The blue dash lines indicate 5% bias while the red dash lines indicate 10% bias.

is observed for the other two sites, although the distributions overrun the 0.95 mark at times, they are always within 10%. The IQR is within the 5% limit in all cases other than L8 B2, where it goes slightly above.

### 4.5 Validation of uncertainty in surface reflectance

Although the assessment of $\sigma_x$ presented above is useful in understanding SIAC performance, we are ultimately more interested in knowing whether BOA reflectance uncertainty values $\sigma_r$ calculated by SIAC are useful in characterising surface reflectance uncertainty. We can do this with the same approach as above, calculating the standardised residual $\epsilon_r$ from Eq. (11) between SIAC reflectance averaged over the RadCalNet AOIs, and the RadCalNet measured reflectance convolved with the appropriate S2/L8 bands. The expectations for this from the discrepancy principle are the same as for $\epsilon_{x_{atm}}$ described above. Fig. 14 shows the distributions of $\epsilon_r$ for the main surface reflectance wavebands.



Figure 14: SIAC surface reflectance uncertainty validation. The red histogram distributions show standardises error distributions from the data and the blue ones the estimated Gaussian distribution from the distributions.





The histograms are quite noisy, particularly for L8, suggesting that the results might be impacted by low sample number. For S2, the mean is very close to 0 for around half of the bands, but can show a positive bias of up to 0.54 (B11). The standard deviation $\sigma$ is less than 1 for all cases, being as low as 0.73 for B11, indicating that we are likely over-estimating the surface reflectance uncertainty by a factor of between 1.04 (B12) and 1.37 (B11) for S2. The L8 analysis shows broadly similar results, with positive bias in the mean of similar magnitude. The values of $\sigma$ however are generally lower, ranging from 0.55 (B06) to 0.75 (B07), suggesting an over-estimation of uncertainty by a factor of between 1.33 and 1.8.

### 4.6 Surface reflectance uncertainty behaviour

The work of Hagolle et al. (2015b) showed that there are patterns we should expect in plots of uncertainty as a function of surface reflectance. Recall that we assume TOA uncertainty $\sigma_y = 0.05y$. We can use the ideas from that paper and examination of the equations here to gain further insights into the factors controlling the uncertainty in surface reflectance and to confirm that SIAC estimates of uncertainty follow the patterns we would expect.

For low AOT and longer wavelengths, the sensitivity to uncertainty in AOT is low, and the term $\Delta_y \sigma_y$ will mostly dominate. For these conditions, $p_b$ and $p_c$ will be very small (see Fig. 2) so $\Delta_y \approx p_a$ from Eq. (B8), so $y \approx r/p_a$ and $\Delta_y \sigma_y \approx 0.05r$. For shorter wavelengths sensitivity to uncertainty in AOT increases. So, even for low AOT, we expect the uncertainty to be more than $0.05r$. For higher AOT and TCWV we expect be significantly higher sensitivity to uncertainty in atmospheric parameters. In this case, we should see uncertainty behaviour similar to Fig 1 in Hagolle et al. (2015b) with a critical value of $r$ for which $\Delta_{H_i^{-1}}$ in Eq. (B7) is zero. For values of $r$ less than or greater than this, the contribution from uncertainty in atmospheric parameters increases, resulting in a 'V'-shape behaviour if we plot $\sigma_r$ as a function of $r$. Fig. 15 shows scatterplots of typical behaviour of this. These are plotted for full S2 scenes for an example of a low AOT case (mean 0.15, ranging from 0.02 to 0.35) and high AOT case (mean 1.1, ranging from 0.9 to 1.25). Results are shown for each waveband, with the colour corresponding to those used in Fig. 2. The turning point reflectance for each band is indicated by a dashed line in that colour.

The turning point feature mainly arises from the AOT component of uncertainty in equation Eq. (B9). We have seen that uncertainty in AOT is expected to be higher for higher AOT (Fig. 5), so for lower AOT this component will be of lower magnitude and the uncertainty being dominated by TOA reflectance uncertainty. For higher AOT, the AOT uncertainty becomes more significant, especially for visible wavebands, and this feature becomes a more dominant part of the uncertainty behaviour, as we would expect.

The black dashed line in the subplots shows the lower boundary of 5% uncertainty that would be expected for a TOA uncertainty of 5%. All values appear on or above this line, providing some confidence in the calculations within SIAC. For low AOT the longer the wavelength, the closer the behaviour directly mimics the TOA relative uncertainty. This arises from the decreasing magnitude of $p_b$ and $p_c$ with wavelength seen in Fig. 2. As these terms become negligible, the TOA and BOA reflectances become more proportionately related, and so when TOA reflectance uncertainty (low AOT), the proportionate TOA uncertainty more closely maps to proportionate BOA uncertainty.





## Reflectance uncertainty vs reflectance

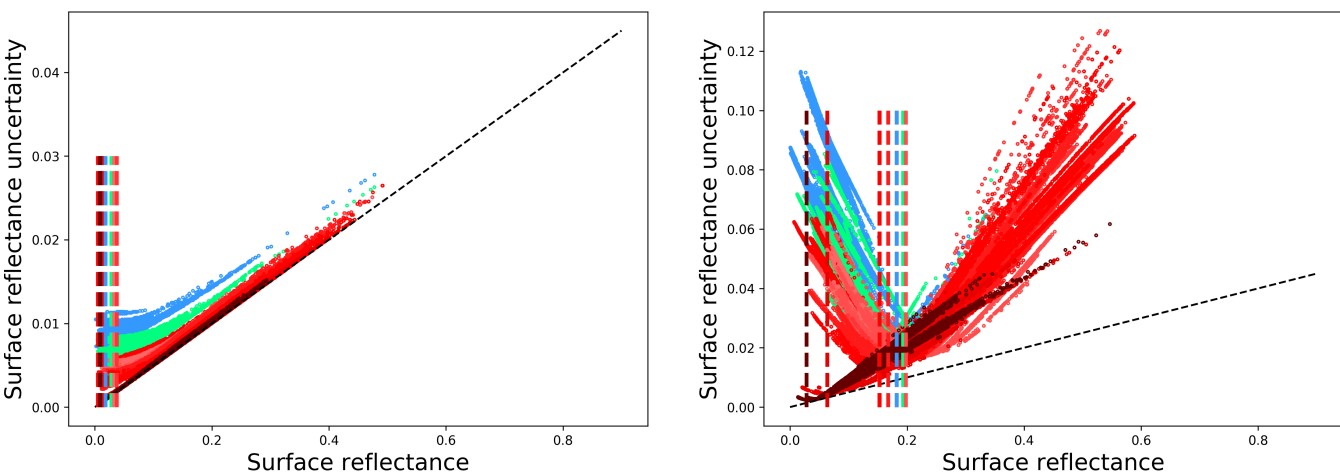

Figure 15: SIAC S2 uncertainty for different bands for low AOT (left) and high AOT(right). The baseline 5% is noted as dashed black line and the minimum of uncertainty of each bands are noted with colored dashed lines.

## 5 Discussion and Conclusions

### 5.1 Contributions of SIAC

Current approaches to atmospheric correction of S2 and L8 data over land use readily available and well-tested atmospheric RT codes such as 6S considered adequate for the task at hand (Vermote and Kotchenova, 2008). Since BRDF effects are not well-sampled, they use the 'simple form' of radiative interaction in Eqs. (3) and (7) allowed by assuming the surface Lambertian. For the most part they proceed by: applying some sort of mask for clouds or other extraneous features; estimating atmospheric state $X$ based in part on observational constraints; and applying $X$ to estimate surface reflectance $r$ via Eq. (7). As we have noted, they do not currently estimate per-pixel uncertainty. Differences in performance between approaches seen in exercises such as the ACIX intercomparisons of (Doxani et al., 2018, 2022) then come as a result of how effective the masks are and how they estimate $X$. We do not attempt to explore the first of these issues in this paper. Issues in the latter can come down to the validity of assumptions that are made, but can also be very dependent on getting a suitable spatial distribution of observational constraints.

SIAC follows these same steps and broad assumptions, but a major feature of the approach is its use of the reliable external operational datastreams available nowadays in support of its estimations. It has other novel features, including accounting for PSF impacts in the scaling from L8 and S2 to MODIS. It is further distinguished by applying a Bayesian framework that is able to weigh up these contributions according to their uncertainty, as well as directly estimate the resultant uncertainty in $X$ and from there, provide a per-pixel estimate of uncertainty in $r$. It is a Data Assimilation approach.



Rather than the more limited extent available for DDV approaches, SIAC uses a direct expectation of surface reflectance from an external coarse resolution source (MCD43) meaning that the keystone observational constraints can supply information to the solution over a wide range of conditions. This means that the solution is, to some extent, reliant on the accuracy of these MODIS reflectance predictions, so we go to some lengths to filter out samples that may not be reliable predictors. In this
first implementation of SIAC, we ignore snow and water pixels as well as some other conditions (Appendix C) which may be over-cautious. The estimate of $X$ in SIAC is based on multiple constraints, so is not in any case entirely dependent on the presence or quality of the observational constraints.

## 5.2  Accuracy assessment

SIAC aims to be CARD4L compliant and to meet threshold requirements for uncertainty. Validation results show that the
method gives accurate (within threshold specifications) retrievals of uncertainty-quantified land surface reflectance, both for S2 and L8 for the most part. Moreover, the surface reflectances for the two sensors are compatible, an important step in using these sensors together for land monitoring applications.

A data assimilation system relies on having well-quantified uncertainties on the constraints used. This is vital in a relative sense for balancing contributions from different sources, but also in an absolute sense for quantifying uncertainty. Unfortu-
nately, none of the constraints we use have a per-pixel estimate of uncertainty to drive the analyses, so instead (Appendix B) we provide reasonable estimates, with references to justify the choice of uncertainty parameters and assess the performance of the uncertainty estimates as part of the validation. In the absence of much observational constraint in a scene or area of a scene, our solution is based mainly on CAMS, so should still provide a good, though less well-spatialised, result. We see this to be borne out in the validation results for AOT and TCWV in Figs. (4) and (6): a small additional percentage of AERONET
samples come within specification in SIAC compared with the CAMS data and the regressions equations remain very similar. What SIAC achieves is a large increase in the correlation coefficient for AOT and a reduction in RMSE suggesting that this localisation is being achieved. When these results are analysed as a function of AOT, the bias A is seen to be very low for AOT up to 0.5 (all comparisons that have more than 11 samples), with U and P lying almost exactly on specification. It is difficult to draw conclusions for AOT values beyond this due to the small sample size. There is some evidence that AOT uncertainty
is slightly higher for L8 than for S2. For TCWV, the RMSE for S2 improves over that of CAMS though it remains the same for L8. For S2, for all comparisons with more than 12 samples, P and U are well within specification, but for L8 this is not the case for TCWV values of less than around 1 $g/cm^2$.

The validation of atmospheric parameters uncertainty in Sect. 4.3 suggest that the uncertainties calculated in SIAC are plausible for the selected values of $\gamma$ according to the discrepancy principle (Sayer et al., 2020). Additionally, we have confirmed
that the uncertainty estimation from SIAC is not strongly affected by the choice of $\gamma$.

The results of comparison between SIAC BOA reflectance and RadCalNet measurements 4.4 suggest that the atmospheric correction is comparable to the atmospheric modelling done with the RadCalNet data driven by observed atmospheric parameters. The number of samples within specification is much higher than expected at the threshold assumed. For most land surface bands and most sites, relative errors are within 5%.





Most of the time, at least over the conditions represented by RadCalNet, we can expect SIAC to estimate surface reflectance $r$ within the threshold specification. Further, the surface reflectances from S2 and L8 are seen to be consistent: the processing has not created artificial biases between the surface reflectance from the two sensors, a feature seen in several papers on the comparison of S2 and L8 surface reflectance (Li et al., 2018; Runge and Grosse, 2019). With the assumed 5% TOA uncertainty, SIAC overestimates surface reflectance uncertainty shown in Sect. 4.5. This suggests that we might reasonably
relax the assumption of 5% uncertainty in TOA reflectance for S2 and L8 towards a 3% value (other than B12) that would be consistent with Lamquin et al. (2018, 2019). The result for L8 is similar to that for S2, but with estimated maximum TOA uncertainty of around 2.5% from this analysis.

Surface reflectance produced by SIAC are of high accuracy and consistent for S2 and L8 as shown in Section 4.4 and 4.5. But we also find that atmospheric correction results derived solely from CAMS atmospheric information over the RadCalNet sites
were seen very similar average behavior (Fig. I1, Fig. I2 and Fig. I3 in Supplementary I). The main impact of the assimilation of observational data then, is a reduction in uncertainty of 10% for S2 visible to near-infrared bands and 5% for S2 SWIR bands with the assumed TOA uncertainty of 3-5%. This is because the TOA uncertainty dominants the total uncertainty budget in the surface reflectance when the AOT is low (mean AOT is around 0.08 over RadCalNet sites). However, we should bear in mind that these sites are not necessarily representative of the challenging environments that might be encountered in global
processing. The RadCalNet sites are located in places with quite homogeneous landscapes and mostly low aerosol loading (RadCalNet, 2018b), which is probably the main reason why there seems little improvement in mean reflectance after assimilation. This suggests that RadCalNet measurements should be used as a minimum test of any atmospheric correction method, rather than a solid evaluation of the accuracy of atmospheric correction, and we ideally need more measurements covering different landscapes with variations of atmospheric states.

## 5.3   Code

The code for SIAC is written in Python, and is released under the GPLv3 open source licence. The code can be obtained from https://github.com/marcyin/SIAC.

*Data availability.*    Data are publicly available through references listed in 3.1.

*Code availability.*    The code for SIAC is written in Python, and is released under the GPLv3 open source licence. The code can be obtained
from https://github.com/marcyin/SIAC.



## Appendix A: Estimation of $R_b(\Lambda_c)$

We need an estimate of BOA BRF at $\Lambda_m$ that we drive from an estimate of BOA reflectance $R_b(\Lambda_c)$ from the MDC43 products. But the observation on the target day may be missing or of low quality. We seek to replace this with a gap-filled estimate based on the product QA flags. Gap filling is achieved with a robust smoother Garcia (2010) having a smoothing factor $s$ of 0.5 (low smoothness). The inverse of bisquare weight $W$ for the target date, given by Garcia (2010) is used to scale relative interpolation uncertainty. Then $\sigma_{r_b}^2$ for each of the 6 bands in Table 3 is given by:

$$\sigma_{r_b,i}^2 = \frac{0.015^2}{W} \frac{\sum_{i=1}^{i=6} \Lambda_i^{-1.6}}{\Lambda_i^{-1.6}} \tag{A1}$$

where the base level uncertainty of 0.015 is the broadband uncertainty in QA=0 retrievals assessed by Wang et al. (2018), the spectral weighting follows Guanter et al. (2007) and is applied to give a conservative estimate of uncertainty in the prediction of $R_b(\Lambda_c)$ and weight lower wavelengths more strongly. Here $\sigma_{r_b,i}^2$ is the value of $\sigma_{r_b}^2$ for band index $i$ with centre wavelength $\Lambda_i$.

## Appendix B: Uncertainty considerations

A data assimilation system combines evidence from different streams by weighting them by their inverse uncertainties. In SIAC, the statistics of the uncertainties is assumed zero-mean Gaussian and thus only characterised by an associated covariance matrix. We review here the sources and values of these uncertainties. The observational and *a priori* constraints for the estimation of $X$ contain inverse covariance matrices $C_{\hat{y}}^{-1}$ and $C_{x_b}^{-1}$ that need definition. We also need to define the uncertainty associated *a posteriori* BOA BRF $R$, $C_r$

### B1   Observational uncertainty $C_{\hat{y}}$

The observational uncertainty in Eq. 2 has three main components that we can call $C_m$, $C_\Lambda$ and $C_G$. The uncertainty associated with the MODIS BRF predictions arises mainly from: (i) uncertainty in the MODIS BRF predictions and the temporal interpolation strategy; and (ii) the spectral transformations. The uncertainty in $r_b(\Lambda_c)$ is given as $\sigma_{r_b}$ in Sect. 2.3.2. The uncertainty from the spectral transformations $C_\Lambda$ is calculated as the RMSE following Supplementary D. The observational uncertainty of the L1C product convolved with estimated ePSF, $C_G^{-1}$, is assumed to be much smaller than $C_m^{-1}$ and ignored in the estimation of $X$. Any other uncertainties arising from the inadequacy of the radiative transfer model are also ignored. We assume $C_{\hat{y}}$ to be a diagonal matrix, so we need only define the variance terms $\sigma_{\hat{y}}^2$.

Following Guanter et al. (2007), we apply an additional spectral weighting over relative wavelength $W_{\Lambda_m} = \Lambda_m^{-1.6}/\overline{\Lambda_m^{-1.6}}$ where $\overline{\Lambda_m^{-1.6}}$ is the mean of central wavelengths of the target sensor in $\Lambda_m$ over all bands raised to the power of -1.6. This has



the effect of giving higher weight to shorter wavelengths to emphasise sensitivity to $AOT$. $C_{\hat{y}}^{-1}$ is taken as a diagonal matrix, with elements:

$$\sigma_{\hat{y}}^{-2} = W_\Lambda \left( \sigma_{r_b}^{-2} + \delta_{\Lambda_m}^{-2} \right) \tag{B1}$$

for each waveband in $\Lambda_m$, defined on $G_c$.

## B2 *a priori* state uncertainty $C_{x_b}^{-1}$

We take the CAMS estimates of atmospheric state at 40 km x 40 km the mean values in $x_b$. Schulz et al. (2020) reports AOT has globally a small bias (-0.04), and an RMSE between forecast and observation of 0.17 versus AERONET match-ups. Zhang

et al. (2020) reports higher values around 0.23 over China, which suggests we should take a more conservative approach in defining the uncertainty, so in SIAC the *a priori* standard deviation for AOT $\sigma_{x_{b(aot)}}$ is a function of AOT, with a minimum threshold. A similar process of comparing CAMS TCWV with AERONET matchups is used to arrive at a relative uncertainty of 0.3 for $TCWV$. $C_{x_b}^{-1}$ in Eq. (5) is assumed a diagonal matrix. We first define standard deviation terms for the variables in $X$:

$$\begin{aligned} \sigma_{x_{b(aot)}} &= \max\left(0.5x_{b(aot)}, 0.02\right) \\ \sigma_{x_{b(tcwv)}} &= 0.3x_{b(tcwv)} \end{aligned} \tag{B2}$$

where $x_{b(aot)}$ and $x_{b(tcwv)}$ are the *a priori* estimates of atmospheric state in $X_b$ on $G_c$. We assume no correlation between uncertainty in AOT and TCWV.

The inverse covariance function is given in Eq. (6) is parameterised by smoothness $\gamma$ and is applied on the grid $G_c$, with the first difference operator $D$ applied across rows and columns[1] The normalisation factor $k^2$ is the mean eigenvalue for

$\left(I + \gamma^2 D^T D\right)^{-1}$ which can be given as (Garcia, 2010):

$$k^2 = \sum_{i=0}^{n} \left[1 + \gamma^2 (2 - 2cos(\frac{i\pi}{n}))\right]^{-1} / n \tag{B3}$$

where $n$ is the number of rows (columns) in the MODIS spatial grid $G_c$ as the term is applied in the row (column) direction.

From the results if a cross-validation study (reported in Supplementary G) we use a $\gamma$ of 5 for S2 and L8 AOT, a value of 5 for S2 TCWV and 0.1 for L8 TCWV.

---

[1]The effect of applying a smoothness constraint in this way is similar to the combined prior and smoothness constraint used in EO-LDAS (Lewis et al., 2012a), GRASP (Dubovik et al., 2011) and Govaerts and Luffarelli (2018), but with an extra normalisation factor $k$ that allows the physical meaning of the inverse correlation and variance terms to be maintained for different degrees of smoothness.





**B3** *a posteriori* state uncertainty $C_x$

Under the assumption that the log-posterior is Gaussian, the mean of the *a posteriori* state $X$ is given by value of $x$ that minimises $J = \mathcal{J}_{obs} + \mathcal{J}_{prior}$, and the *posterior* covariance is approximately given by the inverse of the Hessian at the minimum point (Lewis et al., 2012a):

$$C_x = \left(H'^\top C_{\hat{y}}^{-1} H' + C_{x_b}^{-1}\right)^{-1} \tag{B4}$$

where $H'$ represents the partial derivatives of $\hat{y}$ with respect to $x$. The diagonal of $C_x$ is extracted as the posterior uncertainty for $X$.

**B4 Uncertainty in surface BRF**

Define an augmented state vector $X_a$ on grid $G_m$ containing the atmospheric state variables in $X$ ($AOT$ and $TCWV$) and observations $Y$. Let $\Delta_i$ be the partial derivative of the inverse observation operator $H^{-1}(x_a)$ given in Eq. (7) with respect to variables $i$ in $X_a$, $i \in (AOT, TCWV, y)$:

$$\Delta_i = \frac{\partial H^{-1}(X_a)}{\partial i} \tag{B5}$$

Define $\Delta_{p_j}$ as the partial derivative of the Lambertian coupling term $p_j$ from Eq. (3) for $j \in \{a,b,c\}$ with respect to augmented state variable $i$:

$$\Delta_{p_j} = \frac{\partial p_j}{\partial i} \tag{B6}$$

Then:

$$\Delta_{H_i^{-1}} = -1\left[\frac{-y\Delta_{p_a} + \Delta_{p_c}(p_b - yp_a)^2 + \Delta_{p_b}}{(yp_ap_c - p_b + 1)^2}\right] \tag{B7}$$

The partial derivative of y with respect to TOA reflectance $y$ is:

$$\Delta_y = \frac{p_a}{[p_c(yp_a - p_b) + 1]^2} \tag{B8}$$

So the uncertainty of estimated surface reflectance $\sigma_r^2$ is:

$$\sigma_r^2 = \Delta_{H_{aot}^{-1}}^2 \sigma_{aot}^2 + \Delta_{H_{tcwv}^{-1}}^2 \sigma_{tcwv}^2 + \Delta_y^2 \sigma_y^2 \tag{B9}$$





## Appendix C: Implementation Details

### C0.1 Cloud, shadow, snow and large water body masking

The emphasis in this first version of SIAC is on mapping the land surface, so the L1C TOA S2/L8 data need to have masks applied for areas of cloud, shadow, snow and large water bodies. We describe the approach to this in this section.

Recall from Eq. (3) that estimates of $R_b$ are used to provide sample estimates of $\hat{Y}$ and match against aggregated observations $Y_c$ to estimate $X$ in the observational cost function in Eq. (2). We need to avoid using samples in $R_b$ and $Y$ that are likely to introduce biases in this. An obvious filtering is to avoid pixels that contribute to $Y_c$ that are influenced by extraneous factors such as cloud or cloud shadow. To do this, we calculate masks of these from the TOA reflectance data $Y$ on the original data grid $G_m$, and reject samples from $Y_c$ that would be impacted by these.

For the cloud and cloud shadow mask, we trained an U-NET Convolutional Neural Networks (CNNs) with TensorFlow following Wieland et al. (2019) with training datasets including: the Spatial Procedures for Automated Removal of Cloud and Shadow (SPARCS) dataset (Hughes and Hayes, 2014; USGS, 2016); L8 Biome data (USGS, 2015; Foga et al., 2017); the Sentinel-2 Cloud Mask Catalogue (Francis et al., 2020); and Sentinel-2 reference cloud masks (Baetens and Hagolle, 2018). The overall accuracy obtained is around 0.9, which is similar that of Wieland et al. (2019). A probability of more than 30% is

used the flag cloud, while a 50% threshold is used to for cloud shadow. Finally, a 3 x 3 kernel is repeated 10 times to dilate the cloud and shadow mask and avoid cloud and shadow edge contamination to try to minimise contamination effects.

We also need to consider that some estimates of $R_b$ from the MODIS data may be unreliable. These are likely to be: (i) pixels that are poorly constrained in the MODIS product; and (ii) those undergoing rapid changes during the integration period used in the MODIS product. The first of these should largely have a low QA value if the MODIS sampling data is poor, and be

down-weighted as specified in Appendix B1. However, we also note that we don't expect the BRDF kernels used in the product to be able to deal well with water bodies (they are designed for vegetation canopies) (Schaaf et al., 2021), so should be wary of water pixel predictions. The second will largely be associated with sudden events that have a large impact on reflectance such as snow fall or melting.

In this version of SIAC then, large water bodies are masked out using the ESA global 150 m water products (ESA, 2017)

and snow pixels are masked by $(NDSI > 0.15)\&(NIR > 0.11)\&(GREEN > 0.1)$ from the TOA observations (Zhu and Woodcock, 2012), where $NDSI$ is the Normalised Difference Snow Index (Hall and Riggs, 2011) bands 3 and 11 for S2 and 3 and 6 for L8. Here, NIR refers to band 8 for S2 and 4 for L8. GREEN refers to band 3 for both S2 and L8.

To avoid erroneous spatial features over large water bodies introduced by the excessive extrapolation from distant land pixels, a conservative estimate of atmospheric parameters over water bodies is used, this being the median value of the retrieval

from the rest of the image. This means that SIAC retrievals over water bodies may not be as accurate as over the land surface.

In the retrieval process, if a MODIS pixel on grid $G_c$ contains a medium resolution pixel of grid $G_m$ identified as cloud, shadow, snow and large water bodies in the S2/L8 image, the MODIS pixel will be discarded from the observational constraint. This is quite a conservative approach and the current version of SIAC may have some limitations for or near water bodies and





for snow covered areas. But because of the Bayesian design of the algorithm, even if there is no information available from the
observational data a mostly good estimate of the atmospheric parameters is available from the *a priori* constraint.

## C1 Filtering of MCD43 simulated reflectance $r_b$

The previous masking removes most of the pixels likely to be not reliable for estimates of $R_b$, but we apply a further filtering
step to ensure the robustness of the samples used in the observational constraint. This is based on comparing modelled and
measured BRF using a rough initial estimation of AOT and *a priori* values of water vapour.

This estimate is obtained by deriving a coarse per-pixel (on the $G_c$ grid of the MODIS data) estimate of AOT, then regular-
ising this with a robust smoother to average any noise and remove the influence of any outliers.

A coarse look up table in AOT (AOT 0-2.5 in steps of 0.05) is used to compute $\mathcal{J}_{obs} + \mathcal{J}_{prior}$ for every pixel on grid $G_c$
that passes the initial masking. AOT values corresponding to the minimum cost value for every pixel are used to form an
approximate AOT per-pixel map. This is then filtered with a robust spatial smoother (Garcia, 2010) with a smoothness value
$s$ of 20 to provide a smooth initial estimation of AOT. The robust metric used eliminates the influence of outliers in the AOT
field that may be arise from unreliable values of $R_b$ or other effects.

The smooth AOT estimate then forms part of a rough estimate of atmospheric state $X$ along with other information from
$X_{a_c}$. This is used to generate a first-pass atmospheric correction of observations on the grid $G_c$, $Y_c$, to BOA reflectance $R_c$.
This is compared with the MODIS estimate $R_b$ to derive a residual for each waveband. If the absolute value of this is smaller
than 0.02 for visible bands and smaller than 0.03 for NIR-SWIR bands, the pixel will be used in further processing, otherwise
it is masked, and effectively discarded from consideration.

Although we expect the multiple constraints used in SIAC to provide some degree of robustness to any biases in $R_b$ for
occasional pixels, it is found to be best to filter out gross errors using this approach. The multiple constraints in any case mean
that we do not have to provide measures of $Y_c$ and $\hat{Y}$ for each pixel in $G_c$, and only a sample is required.

## Appendix D: Spectral mapping

### D1 Spectral libraries

Spectral correlation over most natural surfaces suggests that transformations between different spectral domains are possible.
In (Liang, 2001), a set of linear transformations are used to transform from narrowbands (sensor) to broadbands. The linear
relationship is conditional to the spectral library used, but the actual variation of land surface reflectance is much wider than
the variation given by spectral libraries, so there is a need to include realistic spectra outside the spectral libraries. Improving
on the linear transformation, we developed a localised spectral interpolation based on K-nearest neighbours.

The data set used to define these transformations is derived from merging multiple spectral libraries covering the spectral
range the target spectral range re-sampled to 1 nm resolution. To emphasise the importance of vegetation and soils, simulated



vegetation spectra using the PROSAIL model (Jacquemoud et al., 2009) and a soil database were also used. In total, more than 6500 spectra were used. The libraries used are shown in Table D1.

| Library | Target type | Reference | Notes |
|---------|-------------|-----------|-------|
| USGS v7 | Multiple | Pearson et al. (2017) | N/A |
| ASTER | Natural & man-mande materials | Baldridge et al. (2009) | N/A |
| KLUM | Urban | Ilehag et al. (2019) | N/A |
| ICRAF-ISRIC | Soils | Garrity and Bindraban (2004) | Only first 1000 samples used |

Table D1: List of spectral libraries used to define spectral transformations.


Given that the MODIS land bands are designed to capture most of the land surface properties, it should be possible choose library spectra based on how similar these spectra are compared to MODIS surface reflectance, assuming that the spectral library is sufficiently broad. Then, the mean value of the selected spectra are used to predict the S2 and L8 reflectance with their spectral response functions respectively. Although there is a large number of spectra in the spectra library introduced in Sect. D1, it is still not sufficient to cover the vast variations of reflectance seen over the land surface. To deal with this limitation, we split the spectral searching procedure in two parts, visible and infrared spectral region. The spectral selection and comparison between the MODIS reflectance and mean spectral simulated MODIS reflectance are shown in Fig. D1.

We posit that if a spectrum from the library provides a good fit to actual MODIS observations, then this spectrum can be simulated well for any other sensor in the two spectral regions considered. We base this assumption on the strong spectral correlation in those two regions. Based on this assumption, we proceed as follows: rather than using the best fit spectrum to the MODIS surface reflectance, a set of 5 spectra are used to computed the weighted mean using an inverse distance weighting to include some robustness to errors in both the MODIS surface reflectance input and within the spectral library. Other numbers of selected spectra were changed, but it is noted that if this number is large ($> 10$), then it is likely that spectra that are very different from the target spectrum are included, which would be undesirable. Once the mean spectrum is calculated, it is then convolved with the target sensor relative spectral responses (RSRs) to obtain the simulated surface reflectance at $\Lambda_{map}(r_m)$. We use the difference between MODIS measured reflectance and the mean predicted reflectance in the MODIS bands as an indicator of uncertainty.

To test the effectiveness of the proposed spectral mapping method, we simulate the MODIS, S2 and L8 reflectance with individual spectra from the spectral library. Then we use the MODIS simulated reflectance to get K-nearest (K=6 in this case) neighbours spectra from spectral library and discard the spectra used for the computation of MODIS input reflectance, so the remaining spectra are independent from the input reflectance. The mean spectra are computed with weights computed from one divided by the standard Euclidean distance, and the mean spectra simulated reflectance for MODIS, S2 and L8 are computed by convolving with their RSR function. The difference between mean spectral simulated reflectance to the reflectance simulated







Figure D1: Example of spectra selection and comparison between the MCD43 simulated reflectance and mean spectra simulated reflectance.





with an individual spectrum in the spectra library for MODIS is used as a measure of standard error of the mean spectra

simulated reflectance. The result is shown in Fig. D2.

From the spectral mapping results for S2 and L8, we can see that over most of the cases our spectral mapping can simulated both sensors well with high correlation (over 0.99 for all the bands), low RMSE (lower than 0.03 for all bands and 0.015 for the first 5 bands) and no introduced bias introduced. The SWIR band around 2200nm shows the largest dispersion, which is attributed to the large variation in reflectance in this spectral region and the large difference in the band pass functions between

MODIS and S2/L8 shown in Fig. 2.

The standard error estimated from MODIS reflectance provides a reasonably good estimation of the mean spectral estimated reflectance for S2 and L8, since if a large discrepancy between simulations and observations implies that the input reflectance is not well represented in the spectral library.

### D2   Results of spectral mapping

Results of the spectral mapping approach are shown in Fig. D2 over four representative land cover types (vegetation, desert urban and snow). In these plots, the input reflectance is derived from the MODIS MCD43 BRDF products using Eq. (4). The predicted reflectance is in line with the observations, with most of the observations falling within the predictive uncertainty envelope. In the DA system, poor matches to the spectral database will have large uncertainty, and those pixels will have a smaller impact on the inference.

**Appendix E:  Spatial mapping**

Due to the large differences in the spatial resolution between the MODIS $(500\,\mathrm{m})$ and S2/L8 $(10.20\,\mathrm{m}$ and/or $30\,\mathrm{m})$ the measured reflectance values from them can not be directly compared. We model the MODIS data effective PSF, and use this to convolve the high resolution data in order to make it comparable with the MODIS products. Ideally, the MODIS cross track direction PSF is triangular and rectangular in along track direction (Tan et al., 2006; Schowengerdt, 2006), as a result of optical

$PSF_{opt}$, detector $PSF_{det}$, image motion $PSF_{im}$, electronics $PSF_{el}$.

$$PSF_{net}(x,y) = PSF_{opt} * PSF_{det} * PSF_{im} * PSF_{el} \tag{E1}$$

where $*$ is a spatial convolution operator. In the MODIS MCD43 BRDF product, a number of individual observations are inverted together within a temporal window of 16 days. Each of the individual observations has a PSF described as Eq. E1, but the combined product will have an *effective* PSF resulting from the combination of individual measurements with different

angles and scanning geometries. In line with (Kaiser and Schneider, 2008; Duveiller et al., 2011; Mira et al., 2015) we assume that the effective or equivalent PSF ($ePSF$) for the combined product is given by a two-dimensional Gaussian function in along-track (x direction) and across-track (y direction) directions, as shown in Fig. E1:



(a) S2 spectral mapping validation

(b) L8 spectral mapping validation

Figure D2: S2 (up) and L8 (bottom) reflectance predicted by MODIS Aqua selected spectra from spectral library. The uncertainty is shown with 1 $\sigma$ range. The original values are from the direct simulated reflectance by applying S2 and L8 RSR to the individual spectrum in the spectra library.



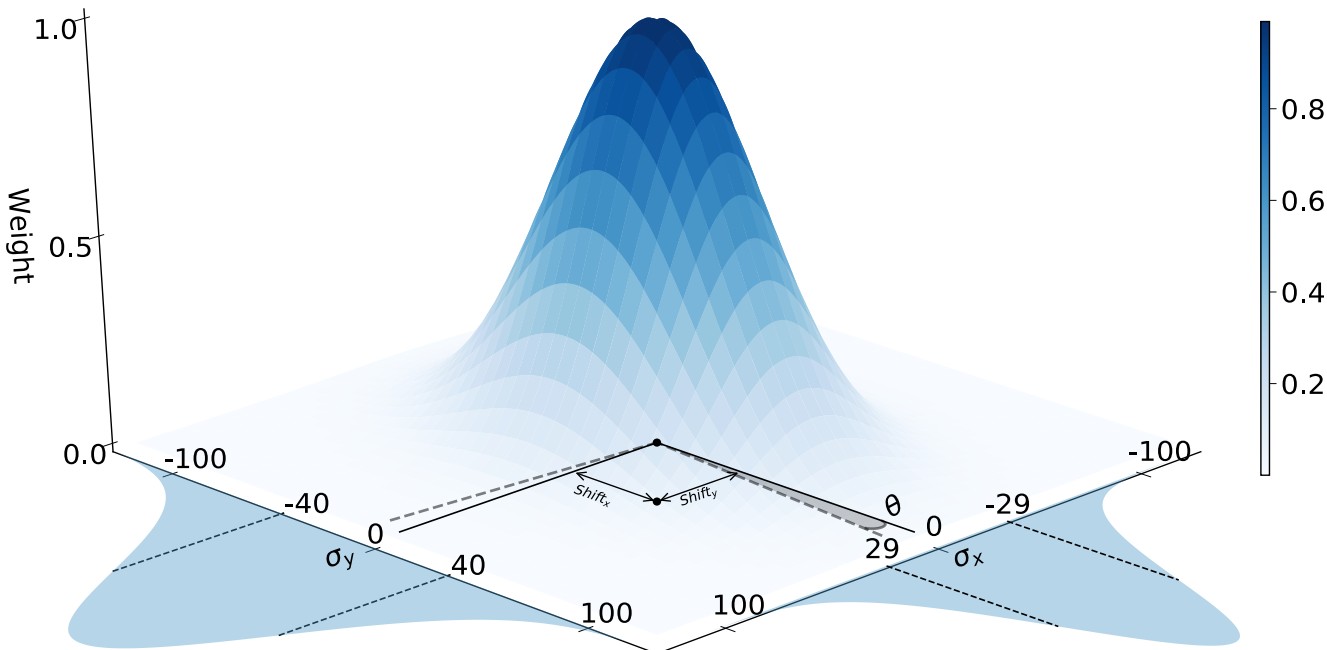

Figure E1: A typical MODIS $ePSF$ on the spatial resolution of S2, *i.e.* unit of 1 represent $10\,\mathrm{m}$ on the X–Y plane, and it follows the same notations as in Equation E2–E4. The shaded area on the two sides represent the Gaussian functions used for x and y directions, with 1 $\sigma$ shown with vertical dashed lines.

$$ePSF(x,y) = \exp\left(-\frac{(x+\Delta_x)^2}{2\sigma_x^2}\right) \cdot exp\left(-\frac{(y+\Delta_y)^2}{2\sigma_y^2}\right) \tag{E2}$$

Where $\sigma_x$ and $\sigma_y$ are the standard deviation of Gaussian function expressed over satellite image pixels unit, $\Delta_x$ and $\Delta_y$
represent the shifts in along and cross directions. According to Duveiller et al. (2011); Capderou (2005) there is also an angular deviation between the satellite orbit and the true north, which is given by:

$$\tan\theta = \frac{\cos i - (1/\kappa)\cos^2\varphi}{\sqrt{\cos^2\varphi - \cos^2 i}} \tag{E3}$$

Where $\theta$ is the angular deviation, $i$ is the inclination angle, $\varphi$ is the latitude and $\kappa$ is the daily recurrence frequency. Then the rotation matrix ($R_\theta$) is:

$$R_\theta = \begin{bmatrix} \cos\theta & -\sin\theta \\ \sin\theta & \cos\theta \end{bmatrix} \tag{E4}$$





We now have an expression that will permit the comparison of the high resolution TOA reflectance with the coarse resolution predictions of surface reflectance obtained from the MCD43 product propagated through the atmosphere. This step is fundamental in defining how the proposed method solves for atmospheric composition, and the equality requires that the high resolution data (S2/L8) are convolved with the $ePSF$. Given the disparity of spatial resolutions, the S2/L8 PSF effect is 690 averaged out during the aggregation and has been neglected in the modelling process.

The spatial convolution is calculated in the frequency domain for efficiency.

### E1    Results of spatial mapping (PSF)

An effective point spread function ($ePSF$) of MODIS MCD43 BRDF product is simulated with a two-dimensional Gaussian function with $\sigma_x$ and $\sigma_y$ controlling the widths of Gaussian in along-track and across-track directions respectively. Shifts in 695 those two directions are also accounted for with estimated $\Delta_x$ and $\Delta_y$ to deal with the geolocation errors in the MCD43 products. Both of the $\sigma$ and $\Delta$ are in the units of pixels for the target sensor, *i.e.* S2 or L8.

We show an example of the PSF modelling in Fig. E2, where we compare the MCD43-predicted reflectance after spectral mapping $r_c(\Lambda(m))$ with S2 TOA reflectance $\hat{y}$ and $\hat{y}(G_c)$ after the spatial mapping at a wavelength around $2200\,\mathrm{nm}$. $r_c(\Lambda(m))$ and $\hat{y}(G_c)$ show broadly similar coarse patterns, but higher spatial details exist $\hat{y}$. Per-pixel level comparison of them shows 700 that $r_c(\Lambda(m))$ and $\hat{y}(G_c)$ has a much stronger correlation, a slope very close to unity and a bias close to zero, indicating that modelling the spatial mismatch is a required step in combining these two datasets.

We have assumed that for a given scene, a single Gaussian PSF is required, in line with the findings of Mira et al. (2015), and we assume further that we can use the PSF derived for the $2200\,\mathrm{nm}$ band for all other bands. At this wavelength, the atmosphere is assumed to be spatially transparent with respect to AOT, and under spectral similarity assumption of the BRF, the results 705 should be similar for other bands (Lyapustin et al., 2011). We illustrate the effect of atmospheric scattering by comparing the $ePSF$-convolved S2 TOA reflectance $\hat{y}(G_c)$ with the MCD43-derived BOA reflectance predictions after spectral mapping $r_c(\Lambda(m))$ in Fig. E3. The pattern shows there are consistent higher atmospheric effects for the shorter wavelengths, that results both in a clear bias due to the important effect of aerosols in the intrinsic path radiance and a slope different to unity (due to the effect of aerosols on upward and downward atmospheric transmission and spherical albedo). For the longer wavelength 710 bands after the NIR plateau, aerosol effects are less important, and the slope is close to unity and the bias close to zero, with the correlation generally being very high, and this also proves the validity of using PSF derived for the $2200\,\mathrm{nm}$ band for all other bands.

After solving for the $ePSF$ parameters over a large number of S2 and L8 scenes globally, we note that some simplifications in the processing are possible. First, we see that the cost function is very flat around the minimum. Fig. E4 shows an example 715 of this: for both S2 and L8, the region around the maximum correlation point has similar values (in excess of 0.98) to the maximum, suggesting that the cost function has limited sensitivities to the $ePSF$ widths when it is close to some optimal values. A second important point is that the width of the $ePSF$ over a large number of scenes tends to be well defined (see Fig. E5 for an example of this). For S2, the median of $\sigma_x$ is around 26 (260 meters) and of $\sigma_y$ is around 34 (340 meters). For L8, these numbers are similar, only that in this case, the number of pixels is three times larger to account for the higher spatial





Figure E2: Comparison between MCD43 simulated surface reflectance after spectral mapping $r_c(\Lambda(m))$ with S2 TOA re-flectance $\hat{y}$ and S2 TOA reflectance after spatial mapping $\hat{y}(G_c)$ on 13/04/2016 S2 50SLH tile. Top row is the $r_c(\Lambda(m))$ and S2 TOA reflectance $\hat{y}$, with the scatter plot between the corresponding pixels (pixel's center geolocation) on the right side. Bottom row is the $r_c(\Lambda(m))$ with $\hat{y}(G_c)$ and their scatter plot.




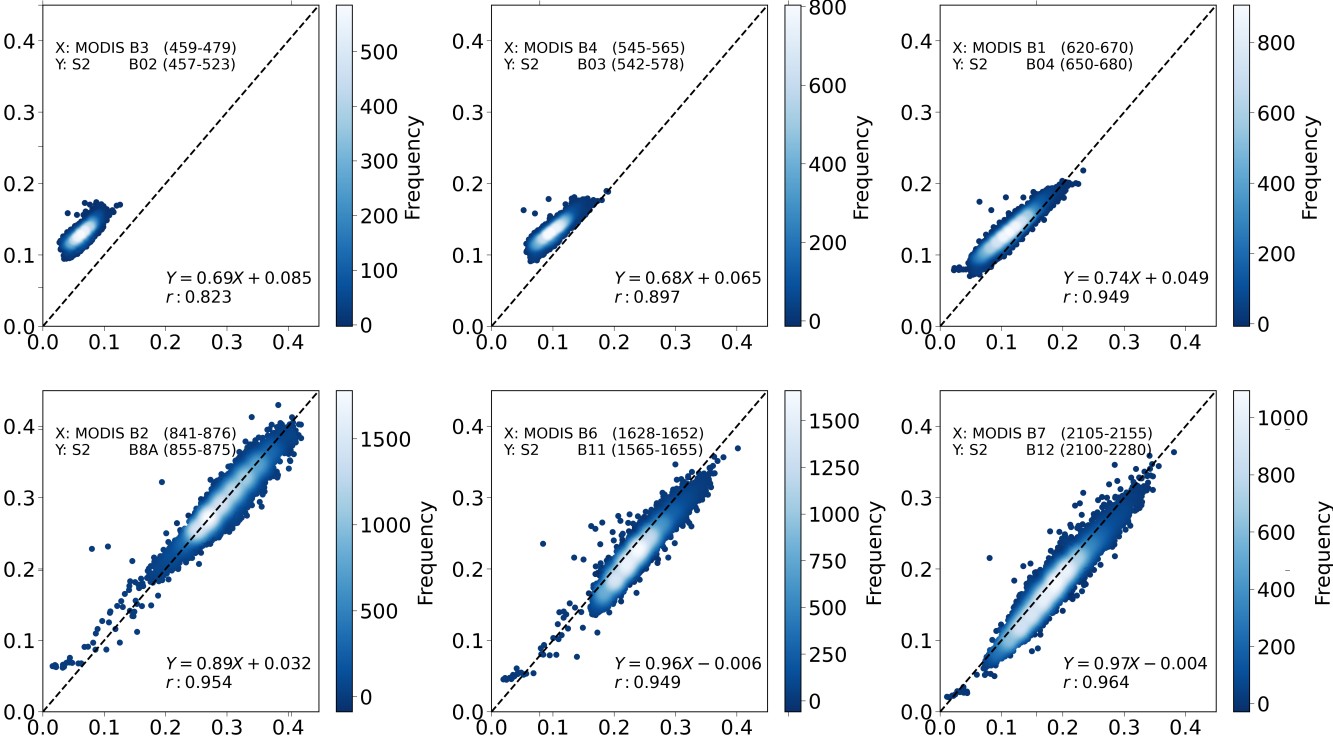

Figure E3: Per-band comparison between the MCD43 simulated surface reflectance $r_c(\Lambda(m))$ and $\hat{y}(G_c)$ at 6 MODIS bands.

resolution of S2. The shift, however, appears more scene-dependent, and also shows a larger influence on the correlation cost function.

The points made above suggest that a fixed value of 260 meters for $\sigma_x$ and 340 meters $\sigma_y$ may be used for all images, but the effect of the pixel shift needs to be inferred on a scene by scene basis. We have not said much of rotation angle $\theta$ (introduced in Eq. E3). In all cases we studied, its value is small (around $\pm 8°$), and, because of the large size of the $ePSF$, its effect can

be effectively compensated by $\Delta_{x,y}$. It is important to note that the aim here is to provide an estimate of an *effective* PSF that allows a comparison between estimates of coarse resolution surface reflectance propagated to TOA and measurements of TOA reflectance convolved with the $ePSF$. To further reduce the computational burden of calculating the $ePSF$ parameters, we have taken the median of $\sigma_x$ and $\sigma_y$ as a reference, assumed the rotation angle of $ePSF$ to be $0°$, and only solved for $\Delta_{x,y}$ on a scene by scene basis.

**Appendix F: Results of Atmospheric parameters inversion and atmospheric correction**

In this Section we illustrate cases of SIAC being used to infer atmospheric composition parameters. Due to S2 and L8 having bands outside from the strong $O_3$ absorption region, $TCO_3$ is taken from CAMS, and only AOT and TCWV are inferred from





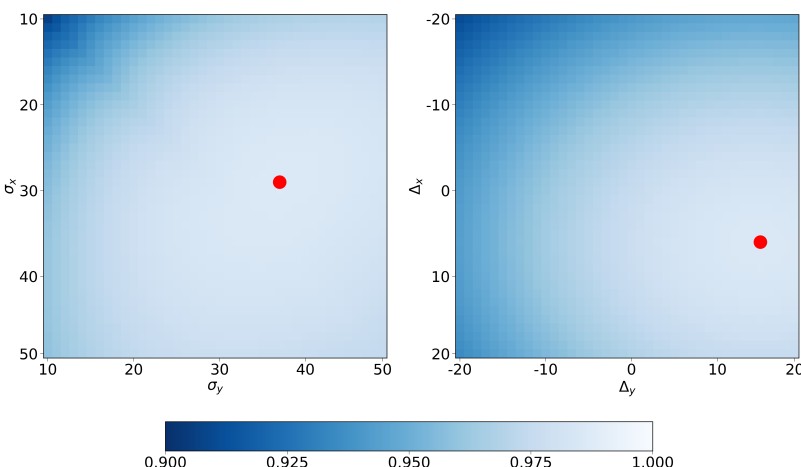

Figure E4: The correlation map between $r_c(\Lambda(m))$ with $\hat{y}(G_c)$ with different $\sigma_x$, $\sigma_y$, $\Delta_x$ and $\Delta_y$ values, in which the red dots represent the largest correlation value's positions.

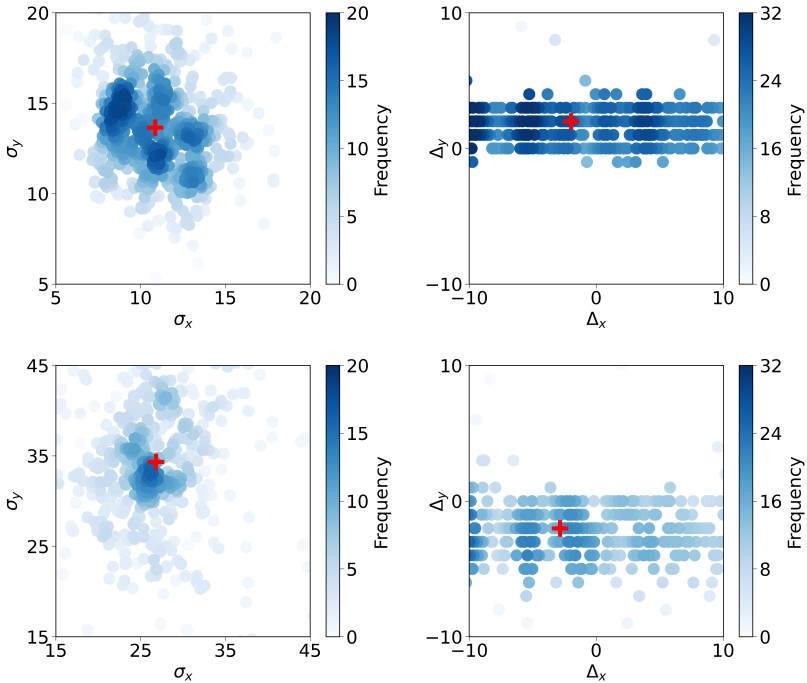

Figure E5: The density scatter plots of solved PSF parameters, $\sigma$(left) and $\Delta$ (right) in x and y direction for L8 (top row) and S2 (bottom row), where the red markers stands for the median of those parameters. The units of the x and y axis are number of pixel.



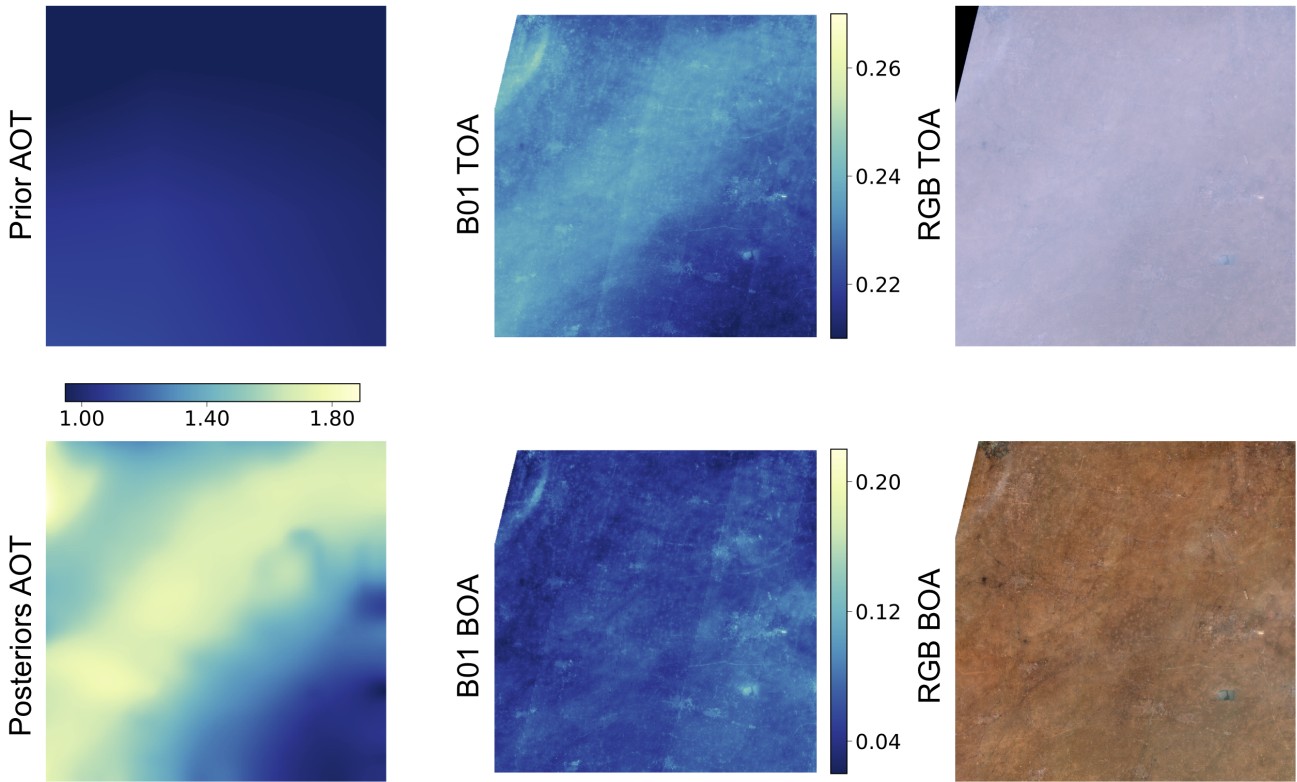

Figure F1: The *prior* and *posterior AOT* over S2 50SMH on 10/02/2016 and their shared colorbar are in the first column. Figures in the second column are band 1 TOA and surface reflectance, while the TOA and BOA RGB images are shown in the third column for the same tile over the same time.

the data. Fig. F1 shows a demonstration of the procedure. Here, an image captured over the North China Plain (tile 50SMH) by S2 on 10/02/2016 has been processed. The CAMS prior mean AOT in Fig. F1 is around 1, and approximately constant over

the scene. The TOA reflectance for band 1 of the S2 sensor (shown in log transformed units to enhance the dynamic range) shows two clear high aerosol stripes. The true colour TOA image shows a very strong atmospheric effect, consistent with the expectation of high AOT. The retrieved AOT (bottom left panel in Fig. F1) shows a marked departure from the prior value. Two very high aerosol stripes are clearly resolved. The result of applying the atmospheric correction results in an important reduction of the atmospheric effects, particularly evident from the BOA true colour composite (bottom right panel of Fig. F1).

Some artefacts are also apparent. In the bottom right corner of the scene, the AOT map reverts to the prior value from CAMS, which results in a poorer correction of the atmospheric effects. This is caused by lack of high quality MCD43 retrievals in this area at this time, which results in the AOT estimate being strongly driven by the prior from CAMS, as well as some spatial





Figure F2: Example of retrieval on S2 data over (a) Amazon ATTO Tower site (S2 Tile 21MTT, 27 Jun. 2019, top panel) and (b) UACJ UNAM OR site (S2 tile 13SCR, 14 Jan. 2019, bottom panel). (Top row of each panel, left to right): AOT prior mean from CAMS, TCWV prior mean from CAMS, blue band TOA reflectance, TOA RGB composite (Bottom row of each panel, left to right): *A poteriori* AOT mean, *A poteriori* TCWV mean, blue band BOA reflectance, BOA RGB composite.



Figure F3: Example of retrieval on S2 data over (a) XiangHe site (S2 tile 50TMK, 13 May 2019, top panel) and (b) Evora site (S2 tile 29SNC, 29 Jul. 2018). (Top row of each panel, left to right): AOT prior mean from CAMS, TCWV prior mean from CAMS, blue band TOA reflectance, TOA RGB composite (Bottom row of each panel, left to right): *A poteriori* AOT mean, *posterior* TCWV mean, blue band BOA reflectance, BOA RGB composite.



diffusive effects from areas where the algorithm performs well. A second artefact is some visible stripes (visible in the middle top and bottom panels). These are caused by the combinations of observations from different detectors (Pahlevan et al., 2017), and have no relationship with the atmospheric correction method. It is also worth noting that when solving for the $ePSF$ parameters for this scene, the optimal linear correlation was only around 0.55, but clearly, the system is still able to produce reasonable results in this challenging environment.

We note that the scene shown in Fig. F1 is a challenging one: at this time of the year, most of the soil is bare, and aerosol loading is very high. Methods that rely on dark dense vegetation (DDV) exploit an empirical relationship between the reflectance around 2200 nm and the blue and red band reflectance. If no vegetation patches are available in the scene (or if their spatial sampling is limited and aerosol spatial dynamics are high), this leads to an inability to obtain a reliable AOT estimate. The deep blue AOT algorithm, used in MODIS aerosol retrieval, has been developed to overcome the shortcoming of DDV method over bright land surface, and the combined products delivers global coverage (Levy et al., 2013; Hsu et al., 2004, 2006). But those two methods have different assumptions and resulting inconsistency in the merged product especially over the transition regions which have comparatively low vegetation cover (Levy et al., 2013).

As a further illustration of the approach on Sentinel 2 data, we show similar visualisations of AOT and TCWV priors, the associated posteriors, TOA and BOA blue band reflectance, as well as TOA and BOA true colour composites for a number of different sites spanning the globe in Fig. F2 (Amazon ATTO Tower and UACJ UNAM OR), Fig. F3 (XiangHe and Evora). While the effect of the atmospheric correction is evident in all these cases, it is important to note that the prior mean is significantly updated when the posterior mean of both AOT and TCWV are calculated. Spatial patterns are clearly visible in all the examples for both parameters, and in many cases, the average value from CAMS changes substantially when the proposed method is deployed. Since the retrieval are made with land pixels only, large water body pixels are filled with median aerosol value from all the land pixels retrieval and the edge between land and water is expected.

## Appendix G: Spatial smoothness parameter estimation

To estimate atmospheric parameters, an estimate of surface reflectance is needed. This estimate is different from the actual surface reflectance and is likely to contain spatial artefacts, which will result in an unrealistic and noisy estimation of atmospheric parameters if an independent pixel level retrieval strategy is used. To counter this, most practical approaches average the estimation of surface reflectance over a fixed size spatial window. Within this window or block, atmospheric composition is assumed constant and inferred (Remer et al., 2009), to reduce the noise in the estimated surface reflectance. This is pragmatic in many ways, and compartmentalises the processing requirements to blocks of that window size. However, the block structure imposed can introduce spurious artefacts if the spatial gradient of atmospheric parameters is large, and fails to estimate atmospheric parameters when no valid targets are found within the specified box.

Within SIAC, the broad scale (40 km) variations of atmospheric parameters are estimated from CAMS. But there are often finer-scale features that may impact our interpretation of surface reflectance, and we wish to be able to resolve these. To this end, we assume an effective resolution of 500 m for atmospheric parameters, with the sub-40 km smoothness constraint



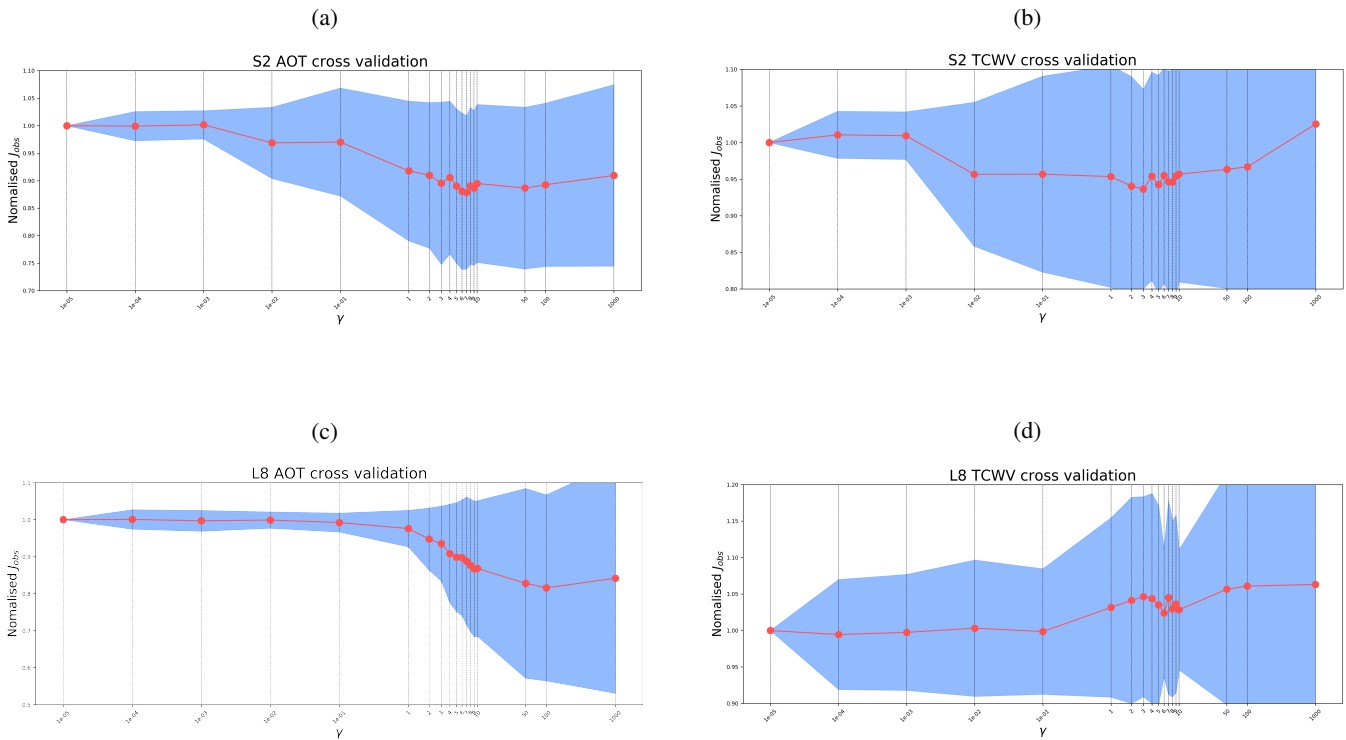

Figure G1: $\gamma$ cross validation for: (a) S2 AOT; (b) S2 TCWV; (c) L8 AOT; and (d) L8 TCWV. The x axis in the subplots is $\gamma$ values ranging from $1e-5$ (essentially, no smoothness constraint) to 1000 (very high smoothness). The y-axis shows $J_{obs}$ normalised by the $J_{obs}$ with a smallest $\gamma$ value of $1e-5$. The red dots show mean normalised $J_{obs}$ over a set of 20 S2/L8 tiles. The blue fill show $\pm$ 1 standard deviation of normalised $J_{obs}$ over the 20 samples.

expressed in 6 and controlled by $\gamma$ allowing for gradual changes in atmospheric parameters in $X$ over the whole of $G_c$. The values in $X$ that we solve for in SIAC are controlled by a weighting of the location and information content of samples in $Y$ and assumed uncertainty of the *a priori* constraint that is, in essence, blurred over $G_c$. The degree of smoothing imposed, and so the resultant (sub-40 km) correlation length of the derived atmospheric parameters, is mainly controlled by $\gamma$. Its squared

inverse, $1/\gamma^2$, a measure of roughness, can also be phrased as the expectation that there is no change at a scale of 500 m (Lewis et al., 2012a). Despite this physical understanding of $\gamma$, it is not straightforward to arrive at a reasonable global estimate of this parameter. We know that too low a value means that $X$ may become over-sensitive to the sample location of the observational constraints and fail to exploit the spatial correlations we know to exist in atmospheric parameters. Too high a value will overly smooth $X$ and it will not be responsive to actual variations over the scene. Fortunately, we know from previous studies (e.g.

Eilers (2003); Lewis et al. (2012a); Eilers et al. (2017)) that results should not be very sensitive to the precise value used, and that there should be quite a wide range of tolerance. In that case, we should select the lowest tolerable value of $\gamma$. In this



context, we can use a cross validation approach over a representative set of scenes to gauge a useful global value. What we would expect to assess from such an experiment is the range of tolerable $\gamma$ values we might use, from which we can select the lowest tolerable value to use within SIAC. In ideal circumstances, we would see a broad, but well-defined minimum in cross
validation cost. An absence of that would suggest a lack of sensitivity of the results to the selection of $\gamma$. Ideally, we would use a consistent $\gamma$ value across different sensors, so that we could have the same assumed degree of smoothing.

We performed a cross validation study using 20 scenes in L8 and in S2, selected to cover a good dynamic range in AOT (0-2). We sampled over $\gamma$ values from $1e-5$ (very low) to 1000 (very high). For each scene, and for each $\gamma$ value, we randomly masked half of the samples in $Y_c$, then solved for $X$ using SIAC. With this $X$, we calculated the observation cost $\mathcal{J}_{obs}$ according to
Eq. (2) for the masked samples in $Y_c$. This gives a measure of observation prediction error (relative to uncertainty) for samples that have atmospheric state interpolated by the correlation function for each scene for given $\gamma$. For very low $\gamma$, the influence of observational samples on predicted atmospheric state at masked locations by will be low, and at these sites, $X$ would effectively be $X_b$. As $\gamma$ increases, the influence of observational samples is higher and we expect to get a better-resolved estimate of $X$, until a point where we are smoothing so much that we lose that resolution. In our experiments, we performed the calculation
of $\mathcal{J}_{obs}$ over 10 random realisations of the masking to obtain an aggregate discrepancy for each scene, normalise this measure by that of the value for the lowest $\gamma$ used, then examine the mean and standard deviation of this measure of cross-validation error over the 20 scenes. The results are shown in Fig. G1.

There is mostly more variation in $\mathcal{J}_{obs}$ over the scenes as $\gamma$ increases, partly due to the normalisation performed, and some sampling noise in the average cost (red points and line), but, as expected, we see a broad minimum region for AOT for S2
and L8 and for TCWV for S2. For L8 TCWV, the result is more complex and we see an increase in cross-validation error for values of $\gamma$ above 0.1. This is because there is very limited sensitivity in the L8 spectral bands to TCWV. For low $\gamma$ (up to 0.1 here) we are effectively seeing the cross-validation error resulting from $X_b$ only. Beyond this point, we are over-smoothing the information in $X_b$, so a global $\gamma$ of 0.1 for L8 TCWV is appropriate. For the other conditions, a compromise value of 5 can be considered as providing a consistent value for use over AOT for both sensors as well as S2 TCWV, although lower values of $\gamma$
for S2 TCWV might also be considered from these results. In the main results section, we use other approaches to verify that these choices are appropriate using other criteria.

## Appendix H: Optimal TOA uncertainty

We show the $\sigma$ of normalised error as a function of TOA reflectance uncertainty in Fig. H1. By changing the TOA reflectance uncertainty to 3%, except for B12 ( 4.5%), we can achieve the optimal uncertainty for the surface reflectance, when the
normalised error distribution have $\sigma$ of 1. This is inline with the validation of L1C products, where the TOA reflectance can achieve 3% (target) of uncertainty for most of the time (Lamquin et al., 2018, 2019). The result for L8 is similar to S2, but it is worth to mention that the optimal uncertainties for L8 TOA are less than 3% for all the bands, with the max of around 2.5%. This could be attributed from fewer co-incident L8 and RadCalNet measurement samples are available due to its longer global revisit time.



(a) Sentinel 2

(b) Landsat 8

Figure H1: Surface reflectance normalised error distribution as a function of TOA reflectance uncertainty. The red dot indicates the point by changing the TOA uncertainty to reach the optimal uncertainty ($\sigma$ of 1 for the normalised error distribution). B6 in for L8 in (b) has $\sigma$ smaller than 1 for all the time, so cannot determine the optimal $\sigma$.



## Appendix I: Improvement over prior correction

We show the correction done by only using the prior values from CAMS predictions in Fig. I1, Fig. I2 and Fig. I3. For most of the time, the CAMS-prior corrected reflectances match the RadCalNet measurements well, but they are worse than the corrections done with SIAC shown in Fig. 10, Fig. 11 and Fig. 12, though the differences are small. This is because those RadCalNet sites are located in places with homogeneous landscape and mostly low aerosol loading (RadCalNet, 2018a). Considering the low sensitivity of surface reflectance to low aerosol loading atmosphere condition, the improvement made on AOT estimation will not improve much on the mean of the estimated surface reflectance. This suggests that the current RadCalNet sites can only be used as a minimum quality check of atmospheric correction or land surface reflectance. More challenging and heterogeneous surface conditions are required for the evaluation of surface reflectance quality. The uncertainty comparison between the prior (CAMS) and posterior (SIAC) corrected surface reflectance in Fig. I4 shows that as the improvements on the uncertainty budget is around 10% for visible to near-infrared bands and 5% for SWIR bands for TOA uncertainty of 5% to 2.5%. The uncertainty improvement result for L8 is much more subtle, this could be attributed from the small sample size. It is also interesting to note that the surface reflectance uncertainty is a function of TOA uncertainty and the proportion of improvements depends on the relative weight between the atmospheric parameters uncertainty and the TOA uncertainty.

*Author contributions.* F.Y. and P.L. conceptualized the study. F.Y. developed the code, prepared the datasets and analysed the results. F.Y. and P.L. wrote the manuscript. J.G.D. suggested experiments, and contributed to the drafting of the manuscript. All authors contributed to editing the paper.

*Competing interests.* The authors declare that they have no conflict of interest.

*Acknowledgements.* The authors would like to acknowledge financial support from the European Union Horizon 2020 research and innovation programme under grant agreement No 687320 MULTIPLY (MULTIscale SENTINEL land surface information retrieval Platform), and from the European Space Agency (ESA) under Contract 4000112388/14/I-NB SEOM SY-4Sci Synergy. F.Y., P.L. and J.G.D. were supported by the Natural Environment Research Council's (NERC) National Centre for Earth Observation (NCEO) (project number 525861). F.Y. and P.L. were supported by Science and Technology Facilities Council (STFC) of UK-Newton Agritech Programme (project number 533651).





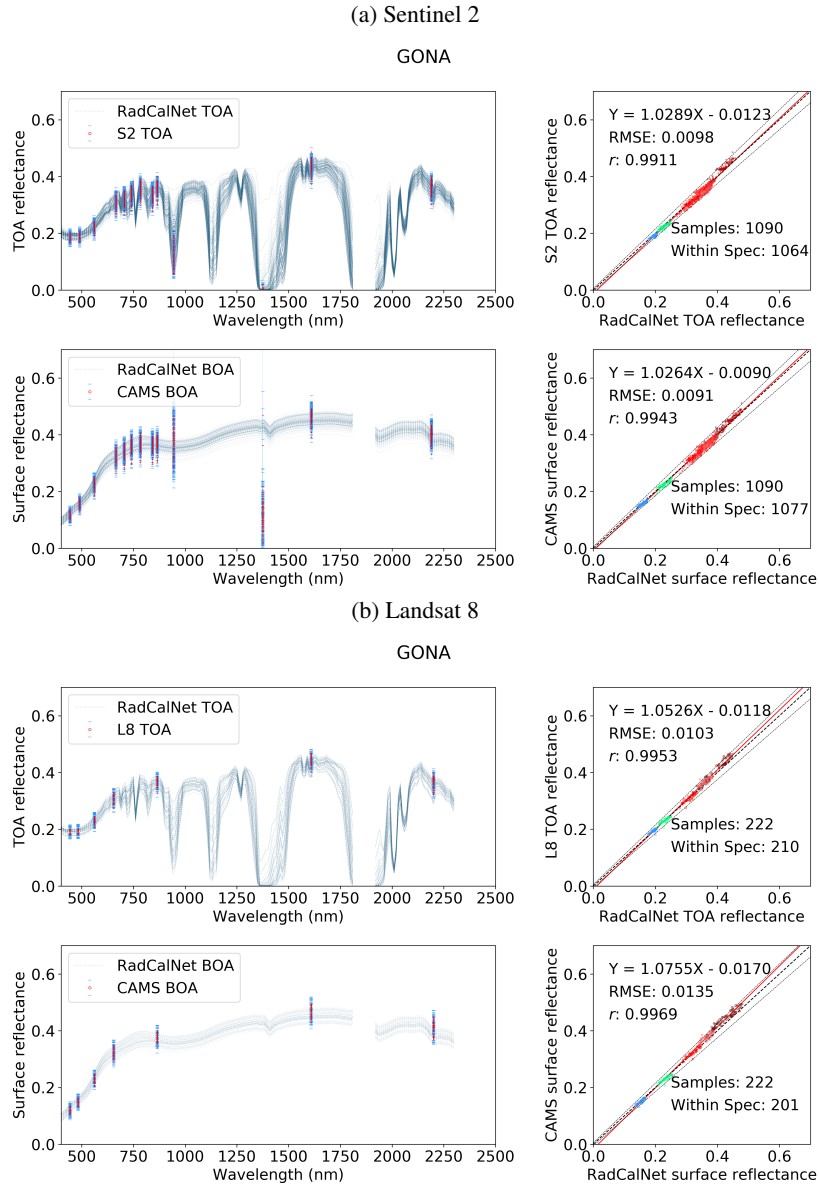

Figure I1: Comparison between the S2 (a) and L8 (b) TOA reflectance and RadCalNet simulated nadir-view TOA reflectance (top row), and the surface reflectance after correction with CAMS prior against RadCalNet nadir-view surface reflectance (bottom row) at Gobabeb. The blue lines at left are different spectra measurement from RadCalNet and the red dot with blue error bars (1 standard deviation) are the TOA or surface and TOA reflectance with uncertainty. The threshold uncertainty is given as black dashed lines in the scatter plots. The regression line is draw as red line and the 1 to 1 reference line is draw as thick black dashed line in the middle.



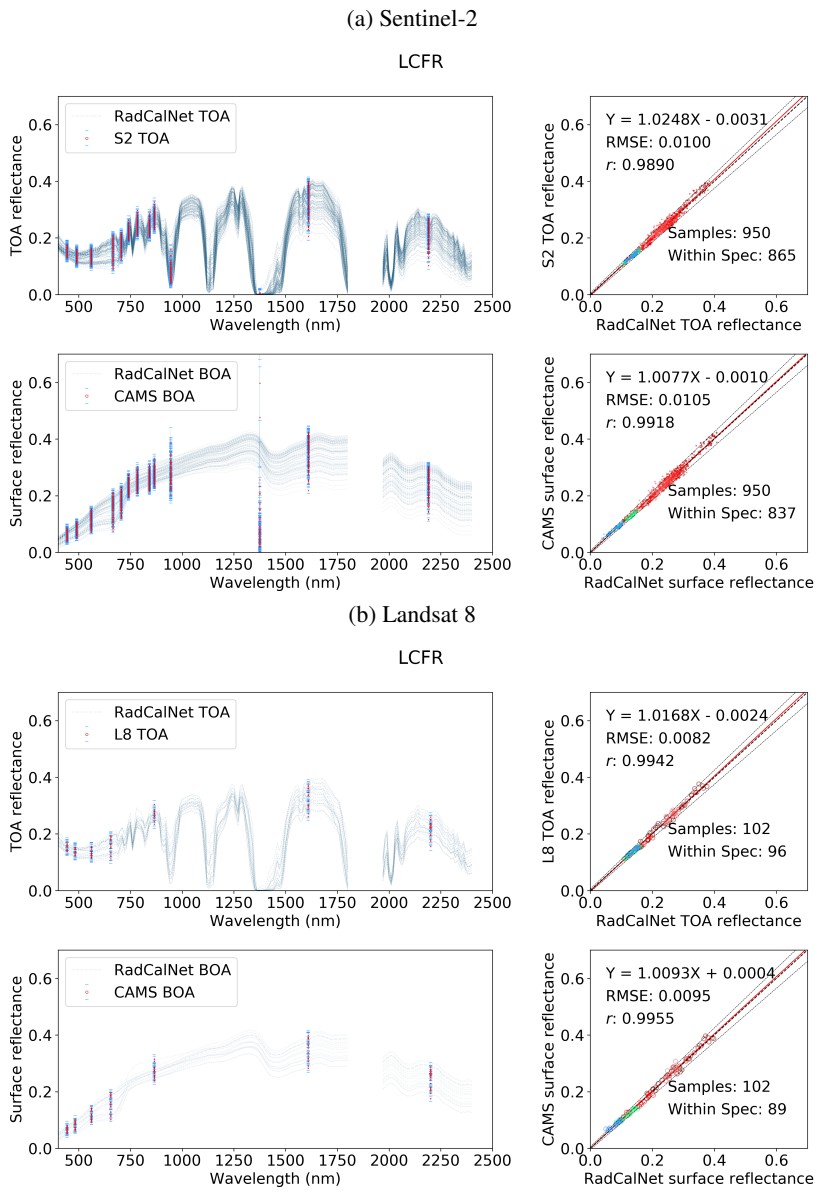

Figure I2: Same as Fig. I1 but for La Crau site.



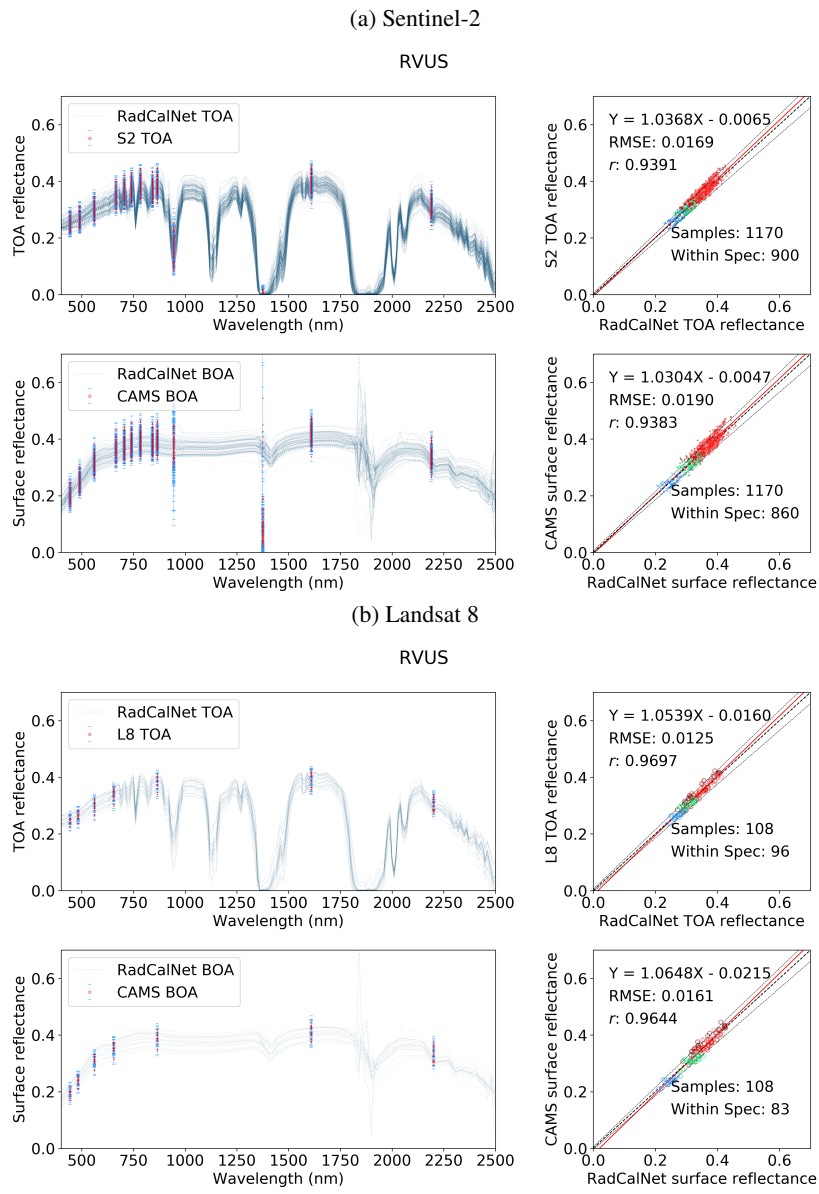

Figure I3: Same as Fig. I1 but for Railroad Valley Playa site.





(a) Sentinel 2

(b) Landsat 8

Figure I4: Surface reflectance uncertainty comparison between CAMS prior corrections and SIAC corrections. The ratio is calculated as CAMS prior corrected surface reflectance uncertainty divided by SIAC surface reflectance uncertainty for different values of TOA uncertainty. The red dot indicates the uncertainty ratio when 5% of TOA reflectance uncertainty used, while the red square indicates the uncertainty ratio when 3% of TOA reflectance uncertainty used.



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
