# Peer review of "BAYESIAN ATMOSPHERIC CORRECTION OVER LAND: SENTINEL-2/MSI AND LANDSAT 8/OLI"

_EGUsphere, 2022_

## Author Comment (AC2)

We thank the review for the insightful comments and suggestions. The responses to the reviewer are in red and modifications to the paper content are in *red and italic with indentation*. The equations, lines and sections in the comments are referred to the updated version of the manuscript.

**General comments:**

Atmospheric correction is an essential part of satellite remote sensing of land surface. Yin et al. describe and evaluate the Sensor Invariant Atmospheric Correction (SIAC) algorithm for atmospheric correction. The existing atmospheric correction methods can be improved and therefore further development of the algorithms is welcome.

SIAC is capable of carrying out atmospheric correction both for Sentinel 2 and Landsat 8 satellite data. Furthermore, SIAC is a Bayesian (statistical) algorithm so it can take advantage of prior information and it produces uncertainty estimates for the surface reflectances produced. The algorithm is tested and validated with ground-based AERONET and RadCalNet data. There are no significant steps taken in new method development in this work, but SIAC combines well existing methods. The results shown in the manuscript show that SIAC is capable of carrying out good quality atmospheric correction.

The selections made in the development of SIAC are mostly well justified and based on previously published literature. It is very good that the authors have shared the codes for others to be used.

My main criticism is in the presentation. The quality of presentation in the manuscript varies. Time to time the text is well written and smooth but quite often the text is difficult to read. The manuscript is quite long and structured so that first SIAC and results are explained in general terms followed by the Discussion and Conclusions, and then all the technical details are mostly included in the Appendices. As this is a method development manuscript, I find this a bit difficult for the reader as it is needed to browse back and forth while reading. Furthermore, the manuscript heavily relies on use of acronyms and symbols. It is good that the main symbols are explained in a table but this also makes the manuscript very slow to read, especially for a person who is not that familiar with the field. I would strongly recommend the authors to think the use of acronyms (even single letter acronyms) and symbols, and possibly shorten and re-organise the manuscript for improved readability. Also, there were some typos in the text so proofreading is recommended. Most of the font sizes in figures are too small and very difficult to read.

This paper describes the theoretical basis, practical implementation, and the validation results, which inevitably makes the paper long and relatively complex. We have decided to put implementation details and further testing in appendices to have a more readable main paper by a wider audience. We have added some text clarifying that the appendices are there to provide detail and additional testing towards the end of Section 2.2:

*In this paper, we present the theoretical underpinnings of the method and major results, relegating details of the implementation and additional results to the appendices.*

Regarding acronyms, the ones used are in common use throughout the literature. Symbols are described in Table 2 and used within the detailed implementation sections of the paper, so we assume the reader is familiar with the paper and overall method at that stage.

We have updated the font size in the figures to improve readability.

**Specific comments:**

l.15 Abstract mentions efficient emulators. However, in the text this was a bit unclear how these were used. Could it be clarified?

l.129 In the abstract this was probably mentioned as "statistical emulation", neural networks are not really statistical emulation.

We have removed the qualifier "statistical" from the abstract and modified the text to be clearer around L129:

Running the model atmospheric model many times is computationally costly, and is often approximated by using e.g, look-up tables. Here, we provide fast surrogate approximations to the full atmospheric model, called emulators (Gómez-Dans et al., 2016). These approximations are based on fully connected artificial neural networks (ANNs) and provide an estimate of the pi terms as a function of the model inputs Xac . Additionally, the Jacobian of the atmospheric model (needed for efficient gradient descent minimisation and for uncertainty propagation) is also approximated by the emulator making use of backpropagation techniques (Hecht-Nielsen, 1992)

I.53 AERONET is based on remote sensing, not in-situ measurements.

Changed to "AERONET estimates" in L203

I.83 ".., we calculate:". "Calculate atmospheric parameter estimation"?

Thanks for pointing this out; it has been changed to:

**SIAC comprises two major steps:**

I.84 The list is difficult to follow.

We modified the list to be a bit more readable.

I.133 "This may cause errors". Can you estimate the significance of these errors

We have changed the text to be clearer:

This choice of aerosol type may cause errors when conditions strongly depart from it, such as situations dominated by urban, maritime or biomass burning conditions. See (Tirelli et al., 2015; Shen et al., 2019) for analysis of the impacts of aerosol types.

We have also added a clarification of this in Sect 5.3 ("Future developments"):

In this paper, the atmospheric composition is set by a model (6S in this case), and by a choice of aerosol optical properties (continental aerosol model). The use of emulators of the RT model makes it easy to change the RT model entirely in the code, or to use a different configuration of the model used. We can also extend the scheme to retrieve independent aerosol species concentrations by both modifying the RT model (and thus extending the number of parameters that go in the inference), and by using data on species distribution available from CAMS and extending the prior to cover these. A similar approach has been implemented in the MAJA processor (Rouquié et al., 2017), which uses the CAMS aerosol species data to define the aerosol types for the atmospheric correction and has found improved atmospheric correction results over deserts. This approach may well be valuable in areas of high dust aerosol loading, or in situations where biomass burning results in an important contribution to aerosol concentrations.

I.169 "Matrix D" was this defined?

It is defined as first-order spatial difference constraint, and the matrix is given in the updated version of the paper (Eq. 7)

I.183 "We assume that mean atmospheric parameters...are constant..." Can you estimate the significance of this assumption?

We have added support for this statement:

We calculate the pa,b,c with the mean atmospheric parameters x and the auxiliary data (Ozone and elevation) at MODIS spatial grid Gc. A linear interpolation is then used to re-sample the pa,b,c to the target sensor grid Gm, which is then used to derive the mean surface reflectance r with Eq. 8. The simple linear interpolation method used to resample the pa,b,c to sub-MODIS scale is justified as atmospheric parameters are known to exhibit much larger correlation lengths (100s of km) (Anderson et al., 2003; Chatterjee et al., 2010).

I.221 Can you give an example of "other artifacts"

We have added a list of likely artifacts:

[...](such as saturation of pixel value, cloud shadow, modelling error from the RadCalNet surface reflectance to TOA reflectance, etc. )

I.223 Why the tolerance of 10% was selected? How this filtering affects the evaluation of the results?

We have added a justification in the text:

5% is used as the target uncertainty of the S2/L8 TOA reflectance and the RadCalNet TOA uncertainty is around 2-5% for non-absorption bands (Wenny and Thome, 2022), which lead us to choose 10% as the threshold to filter out bad samples

I.300 "corrected to R", what is R?

R is surface reflectance (changed in the text)

I.333 Why different TCWV gamma values are used for S2 and L8?

We have pointed out that this is discussed in one of the appendices:

We use y values of of 5 for S2 and L8 AOT, 5 for S2 TCWV and 0.1 for L8 TCWV in SIAC. Cross-validation studies suggest that there is a wide range suitable values for y (Sect. G for additional details on this choice and its implications).

p.23 Figure 13. What are the colors of the bars?

p.26 Figure 15. What are the colors?

We have added a clarification on the boxplots colors

Boxplot colors are the same as colors in Fig. 2.

I.374 Is IQR defined?

It has been updated as the interquartile range (IQR) with the reference added.

I.375 Validation of surface reflectance uncertainty was a bit unclear. Should be clarified.

We have added a summary statement to this paragraph.

*In summary, our tests show that we have a conservative estimate of uncertainty. The overestimation of the variance in surface reflectance is more marked for L8 than S2.*

I.390 In many sentences (especially on page 25) you use "that" & "this" and it is unclear to which word these are referring to

We have updated the text to be clearer in that paragraph.

---

## Author Comment (AC3)

We thank the review for the insightful comments and suggestions. The responses to the reviewer are in red and modifications to the paper content are in *red and italic with indentation*. The equations, lines and sections in the comments are referred to the updated version of the manuscript.

I reviewed the manuscript years ago submitted to another journal + I did look forwarding to the revision which I did not see. Now I saw it in a different journal with significantly improvement in the experimental design and presentation style.

I recommended for publication as the study

- presented a novel way for medium resolution (Landsat-8 and Sentinel-2) satellite data atmospheric correction
- showed the presented method significantly improves the surface reflectance and aerosol property retrieval accuracy
- systematically validated the presented method using Landsat and Sentinel-2 data over globally distributed AERONET

Impressive work and I have to say this is a hard paper to review as there are many techniques used in the study. I appreciated the authors make the codes public available. I like many ideas in the paper but in particular the spatial smoothness prior and the validation of the uncertainty (many geostationary satellite Bayesian AOD estimation output uncertainty without validation). I really hope the authors later (not in this manuscript) can experimentally show that the proposed method is better than LaSRC and Sen2Cor – by running LaSRC and Sen2Cor software on the datasets used in this study (this maybe boring to the authors but I believe it is beneficial to the community and to the authors if the algorithm more convinced). Or simply participate ACIX and hope to see the algorithm will be an unambiguous winner in ACIX.

I have a few comments before the publication of the manuscript

SIAC has been part of the ACIX-II and comparisons will be published soon. The results from SIAC compared in the ACIX-II were submitted more than 2 years ago and many updates used in the current paper had not been implemented by that time.

Major issues

1, The readers may wonder what is the computational cost of the proposed method and the iteration nature of the gradient descent (L-BFGS-B algorithm) may boost the computation time rapidly. Add a paragraph for discussion.

> We added a line at the end of Section 5.1:
>
> > *The entire process (this includes state estimation, surface reflectance estimation, cloud masking and uncertainty quantification) of a single S2 or L8 scene on a standard workstation with a 3.1 GHz i7 CPU and 16 Gb RAM, takes between 20-30 minutes on a single process.*

2, The authors omitted the deep blue bands for both Landsat-8 and Sentinel-2 and the band is most sensitive to aerosol among all the bands, any particular reason for doing so. MODIS does not have this band? use MODIS blue to replace not work ?

Discuss the possible shift from MODIS to VIIRS considering the dying of MODIS.

> We added a comment on these two points in Sect. 5.3 ("Further developments"):
>
> > *SIAC relies on the surface reflectance constrain from the MODIS MCD43 product, but the current MODIS satellites are approaching the end of their mission (Skakun et al., 2018; Xiong and Butler, 2020). VIIRS is designed to produce a continuity data record to MODIS, so it may be possible to use the VIIRS VNP43 product (Justice et al., 2013) as an alternative surface constraint, but this has not been tested. VIIRS land products additionally include bands within the deep blue spectral region (M1, M2), which are more sensitive to the aerosol variation. This information may be used to constrain the S2/L8 deep blue bands to retrieve AOT over bright surfaces (Hsu et al., 2013).*

3, Spatial smoothness is interesting but unclear mostly related to how D matrix is derived. Is the spatial difference applied to the intermediate AOD and WV results derived for each iteration of L-BFGS-B algorithm. If so, Cxb is changing with iterations? The spatial difference is calculated over neighbor 8 pixels or over an entire 40 km window? How the row and column direction are averaged? Simply give an example of deriving D would solve my concern. After the line of 775, "over the whole of Gc" or of sub-40 km? Looks like the sub-40 km is used as a block to solve the equation which means the solution of each sub-40 km is independent to the neighbor sub-40 km block (but I am not sure).

We have tried to clarify the concept prior spatial correlation by rewriting parts of Section 2.4. We hope it is clearer (after L166):

> *This gives a constraint based on a background (a priori) estimate of atmospheric state, Xb. In SIAC, the prior mean comes from the European Centre for Medium-Range Weather Forecasts (ECMWF) CAMS Near-real-time services (Morcrette et al., 2009; Benedetti et al., 2009) for estimates of atmospheric composition parameters AOT at 550 nm, total column water vapour and total column of Ozone in Xb and Xac. These are at a coarse spatial resolution on a 40 km grid, but we need Xb on a 500m grid to match with rb, so the data are interpolated to 500 m resolution.*

> *The role of Cxb is to encode the prior expectation of variance and spatial correlation of the AOT and TCWV fields. AOT and TCWV variances are reported in the CAMS global validation report (see Sect. B2 for a detailed derivation).*

> *We expect the AOT and TCWV fields to have long correlation lengths (Anderson et al., 2003), which result in non-zero off-diagonal elements in Cxb . This spatial correlation structure is defined using a Markov process covariance after Rodgers (2000). This has two free parameters, the variance $\sigma^2$ xb and relative length scale. Since the covariance appears in the constraint Eq. (5) in inverse form, we use a fast approximation this is derived from Rodgers (2000) and implemented as a first-order spatial difference constraint defined in matrix D.*

We have also added a small summary towards the end of the Section describing the effect of these choices, as well as a definition of the D matrix:

> *The matrix D is given as an example for a variable length of n:(7)*

$$
\begin{bmatrix}
-1 & 1 & 0 & 0 & \cdots & 0 \\
0 & -1 & 1 & 0 & \cdots & 0 \\
0 & 0 & -1 & 1 & \cdots & 0 \\
\vdots & \vdots & \vdots & \ddots & \ddots & \vdots \\
0 & 0 & 0 & \cdots & -1 & 1
\end{bmatrix}_{(n-1)\times n}
$$

*Having this prior inverse covariance matrix allows a flow of information from regions in the scene that are well constrained by observations, to areas that are poorly constrained or missing.*

The study need a consistent check of the writing style. For example, lines 102-103.

Line 105, how to treat b09. Is MODIS band 2 reflectance (after D2 transformation) used as a prior for b09 reflectance?

Changed the text to accommodate both comments:

*The output of the spectral mapping provides an estimate of reflectance from 400 to 2400 every 1. This provides a surface reflectance estimation for S2 B09, a band strongly sensitive to TCWV, even though MCD43 does not include this spectral region. Uncertainty from the spectral mapping is explicitly treated in the SIAC framework.*

Line 675, there is another MCD43 PSF estimation paper worth citing (Che et al. 2021).

Che, X., Zhang, H. K., & Liu, J. (2021). Making Landsat 5, 7 and 8 reflectance consistent using MODIS nadir-BRDF adjusted reflectance as reference. Remote Sensing of Environment, 262, 112517.

Added.

Line 130, not quite familiar with the L-BFGS-B algorithm. But does the L-BFGS-B algorithm have its own way to calculate the gradient or the authors really used the back propagation of ANN to calculate ANN (ANN uses an automatic differentiation method) to feed into L-BFGS-B algorithm. Or back prorogation gradient is only used in B4 for RBF uncertainty but not used in L-BFGS-B?

We have added some text to describe the use of emulators as providers of the Jacobian of the cost function:

*The cost function and its partial derivatives exploit the ability of the emulators to provide accurate approximations to the atmospheric RT model and its partial derivatives (Gómez-Dans et al., 2016).*

Line 140, the LaSRC cannot be considered as DDV method anymore as each single pixel on the global have a visible/SWIR ratio (not just DDV pixels) see Vermote et al. 2016.

Yes, LaSRC is a variant of it, but strictly speaking it cannot be treated as DDV. So, we have removed it from the list.

Line 155, the framework can tolerate incomplete coverage of $Y^$, how? If incomplete $Y^$, then the incomplete inversion of aod and wv will be spatially interpolated? Or get value using spatial smooth prior.

This is the role of the spatial correlation in the prior pdf (Sect 2.4). We have already expanded on this as a response to another comment, and hope it is clearer now.

Line 200, "a estimates" – grammar

Changed

Line 260, how the $\sigma x$aero is set

We have added a reference to Sect. 3.2, where these magnitudes are defined.

Line 310, which TOA band, the blue band should definitely greater than 10% mostly

We disagree with this comment. The filter is applied to each band of the 6 compared bands. Those two sensors agree better than 5% for most of the time even the blue and deep blue bands, please refer to paper mentioned in the previous line (309). Another paper by Helder, et al., (2020): https://doi.org/10.3390/rs10091340

Line 330 we are already in Section 4.3

Removed this line

Please label and re-arrange figures 10-12 correctly. Some are clearly surface but labelled as TOA. Check the labels of the plots. Rearrange them so that the TOA spectra and TOA plot are at the same row.

After checking figures 10-12 (11-13 in the new version), they are in the correct order. You may say that the surface reflectance cannot have a better statistic than the TOA, but it can happen as the surface reflectance has a better contrast and stretches the reflectance values to a wider range than the TOA reflectance. We have added some clarification on this:

*The fact that sometimes the BOA statistics are better than the TOA is probably due to the better dynamic range of the surface reflectance signal compared to the TOA one.*

Line 470, yes, in many Landsat calibration literature, the calibration accuracy is 3%. Should this linked to appendix H?

Yes, this has been linked to Appendix H.

Line 580. There is no Schaff 2021 in reference list

It is in Line 1110.

---

## Author Response (AR2)

**Authors' Response to Reviews of**

**BAYESIAN ATMOSPHERIC CORRECTION OVER LAND: SENTINEL-2/MSI AND LANDSAT 8/OLI**

Feng Yin, Philip E Lewis, Jose L Gómez-Dans

*Geoscientific Model Development,* `https://doi.org/10.5194/egusphere-2022-170`
* * *
**RC:** *Reviewers' Comment*,     AR: Authors' Response,     ☐ Manuscript Text

**1. Reviewer #2**

**1.1. Concern #1**

**RC:** *Thanks for the authors' response and addressing for most of my concerns. I still could not follow how the spatial smoothness prior works.*

*I am curious what is the spatial support in solving the Bayesian Equation (1). If there is no spatial smoothness, the Bayesian Equation can be solved at each Landsat/Setinel-2 pixel as neighbor Landsat/Setinel-2 pixels are independent. However, with spatial smoothness the Bayesian equation cannot be solved for each single pixel anymore, and a cluster of pixels will be put tougher so that D can be applied (I assume D matrix is applied on the water vapor and aerosol images). It looks like the authors choose to solve all Landsat/S2 pixels within a MODIS pixel together, i.e., D is applied all L8/S2 pixels within a MODIS pixel. In another word, the solution induced by the spatial smoothness is only applied within a MODIS pixel. The L8/S2 pixels coming from neighbor MODIS pixels are not constrained by the spatial smoothness even the two L8/S2 may be close to each other at the MODIS pixel boundaries (of course the neighbor MODIS pixels may very likely use the same prior as the prior is defined at 40 km resolution).*

**AR:** We thank the reviewer for the insightful comments and suggestions. The spatial resolution for solving equation 1 is at Gc, which is also the spatial resolution used to define the D matrix. All the inputs to the solver have been processed to Gc, i.e. MODIS spatial resolution, and we solve the atmospheric parameters for the whole S2/L8 tile at grid Gc. Therefore, the D matrix is applied to the neighbouring MODIS pixels rather than the S2/L8 pixels within a MODIS pixel, as the S2/L8 TOA observations have been aggregated to Gc during the PSF modelling procedure.

We have added text in line 116-117 to clarify the spatial resolution of TOA observation (at Gc) after PSF modelling:

> The output through the ePSF modelling provide us with the TOA observations on grid $G_c$, i.e. MODIS grid.

**AR:** And in line 185-187 to clarify the spatial resolution of D matrix:

> The  TOA reflectance observation, modelled TOA reflectance and prior information on the atmospheric parameters are processed to $G_c$ through the procedures

described above. Thus, the D matrix is also defined at $G_c$, i.e. 500 m to provide spatial constraint for the atmospheric parameters. For a variable length of n, the matrix $D$ is given as

**1.2. Minor points**

**RC:** *Line 80: "with an associated sub-40 km spatial correlation constraint" – But if my above statement is correct the spatial correlation is only on 500 m.*

**AR:** Yes, the spatial correlation is at 500 m and we have removed the sub-40 km and clarified the spatial constraint is for atmospheric parameters (AOT, TCWV) in line 81:

> Our approach uses *a priori* constraints in the form of a coarse resolution (500m) spectral BRDF dataset from MODIS (Schaaf and Wang, 2015) to provide *sample* land surface reflectance estimates, as well as a very coarse resolution (40 km) estimate of atmospheric composition from the CAMS near-real-time global assimilation and forecasting system (Morcrette et al., 2009; Benedetti et al., 2009), with an associated  spatial correlation constraint for the atmospheric parameters.

**RC:** *Table 2, Should X and Xb defined for L8/S2 resolution rather than MODIS resolution.*

**AR:** As clarified above, all the inputs to the solver are at Gc, MODIS resolution

**RC:** *After Equation B3, "where n is the number of rows (columns) in the MODIS spatial grid Gc as the term is applied in the row (column) direction". This is where my interpretation coming from since n is the number of L8/S2 pixels in a MODIS grid Gc. It implies the D with n\*n size is applied to all L8/S2 pixels in a MODIS grid.*

**AR:** The original text causes the confusion and the description has been changed in Line 587-588:

> where $n$ is the number of rows (columns)  of pixels within the S2/L8 spatial extent at the MODIS spatial grid $G_c$ as the term is applied in the row (column) direction.

**RC:** *Line 855, "expressed in 6 and controlled by allowing for gradual changes in atmospheric parameters in X over the whole of Gc". This is also where my interpretation coming from since the parameters X over Gc has gradual changes and consequently X should be defined at L8/S2 resolution (Gm).*

**AR:** The preposition 'in' has caused the confusion for the reviewer, and we have changed Line 820-823:

> Within SIAC, the broad scale (40 km) variations of atmospheric parameters are estimated from CAMS. But there are often finer-scale features that may impact our interpretation of surface reflectance, and we wish to be able to resolve these. To this end, we assume an effective resolution of $G_c$ (500 m) for atmospheric parameters, with the  smoothness constraint expressed in Eq. 6 and controlled by $\gamma$ . This spatial constraint allows for gradual changes  in atmospheric parameters $X$  at the spatial resolution of $G_c$ over the S2/L8 image spatial extent. The values in $X$ that we solve for in SIAC are controlled by a weighting of the location and information content of samples in $Y$ and assumed uncertainty of the *a priori* constraint that is, in essence, blurred over $G_c$. The degree of smoothing imposed, and so the resultant  correlation length of the derived atmospheric parameters, is mainly controlled by $\gamma$. Its squared inverse,

> $1/\gamma^2$, a measure of roughness, can also be phrased as the expectation that there is no change at a scale of 500 m (Lewis et al., 2012).

**RC:** *If I am wrong, please find a place to clarify, better in 2.2 and 2.4 both.*

AR:  Clarification have been made in section 2.2 and 2.4 stated above.

**References**

A. Benedetti, J.-J. Morcrette, O. Boucher, A. Dethof, R. Engelen, M. Fisher, H. Flentje, N. Huneeus, L. Jones, J. Kaiser, et al. Aerosol analysis and forecast in the european centre for medium-range weather forecasts integrated forecast system: 2. data assimilation. *Journal of Geophysical Research: Atmospheres*, 114(D13), 2009.

P. Lewis, J. Gómez-Dans, T. Kaminski, J. Settle, T. Quaife, N. Gobron, J. Styles, and M. Berger. An Earth Observation Land Data Assimilation System (EO-LDAS). *Remote Sensing of Environment*, 120(0): 219–235, 15 May 2012. ISSN 0034-4257. .

J.-J. Morcrette, O. Boucher, L. Jones, D. Salmond, P. Bechtold, A. Beljaars, A. Benedetti, A. Bonet, J. Kaiser, M. Razinger, et al. Aerosol analysis and forecast in the european centre for medium-range weather forecasts integrated forecast system: Forward modeling. *Journal of Geophysical Research: Atmospheres*, 114(D6), 2009.

C. Schaaf and Z. Wang. MCD43A4 MODIS/terra+aqua BRDF/albedo nadir BRDF adjusted refectance daily l3 global - 500m v006. `https://doi.org/10.5067/{MODIS}/MCD43A4.006`, 2015. URL `https://doi.org/10.5067/{MODIS}/MCD43A4.006`. Accessed: 3-3-2020.

**Authors' Response to Reviews of**

**BAYESIAN ATMOSPHERIC CORRECTION OVER LAND: SENTINEL-2/MSI AND LANDSAT 8/OLI**

Feng Yin, Philip E Lewis, Jose L Gómez-Dans

*Geoscientific Model Development,* `https://doi.org/10.5194/egusphere-2022-170`
* * *
**RC:** *Reviewers' Comment*,     AR: Authors' Response,     ☐ Manuscript Text

**1. Reviewer #3**

**1.1. Major Concern #1**

**RC:** *The manuscript of Feng Yin proposes a method for atmospheric correction for radiometers with a typical spatial resolution of few tenths of meters relying on prior information determined at much lower spatial resolution (few hundredths of meters). The proposed method relies on a Bayesian approach. Though the title of the manuscript "BAYESIAN ATMOSPHERIC CORRECTION OVER LAND: SENTINEL-2/MSI AND LANDSAT 8/OLI" is very appealing, this research work misses delivering the expected Analysis Ready Data outcome, essentially due to several unjustified assumptions and approximations. As stated in the abstract, mitigating the impact of atmospheric effects on optical remote sensing data is critical for monitoring intrinsic land processes. In the context of delivering Analysis Ready Data, intrinsic land processes mean that no additional variables are needed to analyse the delivered surface reflectance. Despite the apparent complexity of the proposed SIAC Bayesian approach, the delivered geophysical variable, namely the bottom of atmosphere Bidirectional Reflectance Factor (factor and not function as stated in line 29) is erroneously presented as Analysis Ready Data. It still depends on the actual state of the atmosphere! [...] Unfortunately, I would not recommend the publication of this manuscript as long as it relies on erroneous assumptions.*

**AR:** We want to thank the reviewer for his/her time, but we note that this is only a very partial review of the manuscript. As a general comment, the reviewer claims that higher order effects need to be accounted for in order to produce a useful Analysis Ready Data (ARD) product. We note that this is not a requirement of the CEOS ARD for Land (CARD4L) (CEOS, 2021b). The assumption of a Lambertian surface is made in most of the atmospheric correction schemes that take part on the Atmospheric Intercomparison Exercise (ACIX) (Doxani et al., 2018, 2022), and a number of surface reflectance products adopting this assumption are accepted as CEOS assessed ARD products (CEOS, 2021a). We have also quoted the last sentence, which we think is inappropriate: we have made an assumption that is commonly used, and have tried to quantify its impact. This does not make it "erroneous"

**1.2. Major concern #2**

**RC:** *In line 60, it is written "We wish to estimate the probability distribution function (PDF) of BOA spectral BRF, R with illumination and viewing vectors $\Omega_s$, $\Omega_v$ respectively". The Bidirectional Reflectance Factor is indeed an intrinsic property of the surface that depends on $\Omega_s$, $\Omega_v$ The BRF should be estimated at the surface to achieve this objective, removing all possible atmospheric effects as written in line 94. In line 120 Equation (3), it is however assumed that the surface reflectance rb represents a "Lambertian reflectance" which is equivalent to assuming that the sky radiation is perfectly diffuse which is true only when the sky*

*is overcast. rb is however estimated with Equation (4) which provides the BRF for the $\Omega_s$, $\Omega_v$ geometry. It is therefore not a Lambertian Equivalent Reflectance as assumed in Equation 3. The relationship between rb expressed in Equations (3) and (4) is thus inconsistent. Assuming that rb in Equation (3) can be estimated with Equation (4) is strictly correct for the direct contribution only. It is inconsistent with the definition of the terms representing the total (direct and diffuse) downwelling and upwelling atmospheric transmittance. Consequently, the retrieved surface reflectance still depends on the actual atmospheric state (diffuse contribution). Hence, as long as this issue is not properly solved, the proposed approach will not work for the delivery of Analysis Ready Data. It is a major flaw in the proposed approach that should be corrected, and they are elegant ways to do so!*

AR:  Errors caused by higher order effects, such as BRDF coupling, only become dominant when there is high aerosol loading, large sun/view zenith angle and strong surface anisotropy. To account for those higher order effects in the atmospheric correction, knowledge on the surface BRDF is required, which is not available for medium resolution (10 m - 60 m) data. This is also likely to be the reason why most of the current methods assume a Lambertian surface. One option we have explored and will present in future work is to consider the coupling at a coarser spatial resolution from the ancillary MODIS data, but that is difficult to validate with the data we present in this paper. We stress again that the CARD4L, we aiming to produce, does not specify the requirements on the correction higher order effects. However, the reviewer rejected our paper with his subjective view on the ARD and ignored the community agreed standards.

To further clarify the potential errors caused by the Lambertian assumption, we have added some quantification of the likely uncertainties introduced by this assumption based on the literature in Section 2.2 as follows:

> We assume a Lambertian surface in the atmospheric correction process. The relative errors caused by this Lambertian assumption on the surface reflectance is 3-12% in the visible bands and 0.7–5.0% in the near-infrared bands. Its effect on the NDVI analysis is around 1% and less than 1% for albedo (Franch et al., 2013), which is within 5% accuracy requirement on albedo by the Global Climate Observing System (GCOS) (GCOS, 2019). This Lambertian assumption is also widely used to produce the surface reflectance for MODIS (Franch et al., 2013), Landsat (Vermote et al., 2016) and S2 (ESA, 2021) , where the Landsat and S2 surface reflectance products have been accepted as the CEOS assessed ARD products (CEOS, 2021a).

**1.3. Minor points**

RC:  *In lines 19 and 20, R defines a correlation coefficient.*

AR:  $R$ has been changed to correlation in the abstract. Thanks.

> SIAC is demonstrated for a set of global S2 and L8 images covering AERONET and RadCalNet sites. AOT retrievals show a very high correlation to AERONET estimates ($\bcancel{R}$ correlation coefficient around 0.86,   RMSE of 0.07 for both sensors), although with a small bias in AOT. TCWV is accurately retrieved from both sensors $(\bcancel{R > 0.96, RMSE < 0.32 \ g/cm^2})$(correlation coefficient over 0.96, $RMSE < 0.32 \ g/cm^2$). Comparisons with *in situ* surface reflectance measurements from the RadCalNet network show that SIAC provides accurate estimates of surface reflectance across the entire spectrum, with $RMSE$ mismatches with the reference data between 0.01 and 0.02 in units of reflectance, for both S2 and L8. For near-simultaneous S2 and L8 acquisitions, there is a very tight relationship ($\bcancel{R > 0.95}$ correlation coefficient over 0.95 for all common bands) between surface reflectance from both sensors, with negligible biases. Uncertainty estimates are assessed through

discrepancy analysis and found to provide viable estimates for AOT and TCWV. For surface reflectance, they give conservative estimates of uncertainty, suggesting that a lower estimate of TOA reflectance uncertainty might be appropriate.

**RC:** *In Table 1, the symbol r is not defined.*

AR: Added definition in the table caption:

Threshold uncertainty specifications for Aerosol Optical Thickness (AOT), total column of water vapour (TCWV) and BOA BRF (r) used in this paper.

**RC:** *In Table 2, R is defined as R N(r,Cr) as the a posteriori PDF of the BRF but the symbols r and Cr are not defined.*
*In Table 2, Rb is defined as Rb (rb,Cb) as the a priori PDF of the BRF but the symbols rb and Cb are not defined*

AR: Clarification on C has been added to the Table caption:

Main symbols used in the paper. $C_*$ represents the covariance matrix part of the PDF for parameter $*$.

**RC:** *In line 200, r is defined as the mean surface reflectance.*

AR: Yes, r is the mean surface reflectance estimate under Lambertian assumption

**References**

CEOS. CEOS Analysis Ready Data, Jul 2021a. URL `https://ceos.org/ard`. [Online; accessed 21. Sep. 2021].

CEOS. Analysis Ready Data For Land, 2021b. URL `https://ceos.org/ard/files/PFS/SR/v5.0/CARD4L_Product_Family_Specification_Surface_Reflectance-v5.0.pdf`.

G. Doxani, E. Vermote, J.-C. Roger, F. Gascon, S. Adriaensen, D. Frantz, O. Hagolle, A. Hollstein, G. Kirches, F. Li, J. Louis, A. Mangin, N. Pahlevan, B. Pflug, and Q. Vanhellemont. Atmospheric correction inter-comparison exercise. *Remote Sensing*, 10(3):352, feb 2018. . URL `https://doi.org/10.3390/rs10020352`.

G. Doxani, E. Vermote, J.-C. Roger, S. Skakun, F. Gascon, A. Collison, L. D. Keukelaere, C. Desjardins, D. Frantz, O. Hagolle, M. Kim, , J. Louis, F. Pacifici, B. Pflug, H. Poilvé, D. Ramon, R. Richter, and F. Yin. Atmospheric correction inter-comparison exercise (acix ii land): an atmospheric correction processors assessment for landsat 8 and sentinel-2 over land. *Remote Sensing of Environment*, 2022.

ESA. S2 mpc level-2a algorithm theoretical basis document, 2021. URL `https://step.esa.int/thirdparties/sen2cor/2.10.0/docs/S2-PDGS-MPC-L2A-ATBD-V2.10.0.pdf`.

B. Franch, E. Vermote, J. Sobrino, and E. Fédèle. Analysis of directional effects on atmospheric correction. *Remote Sensing of Environment*, 128:276–288, Jan. 2013. . URL `https://doi.org/10.1016/j.rse.2012.10.018`.

GCOS. Albedo essential climate variable (ecv) factsheet, 2019. URL `https://gcos.wmo.int/en/essential-climate-variables/albedo`. [Accessed 12-Sep-2022].

E. Vermote, C. Justice, M. Claverie, and B. Franch. Preliminary analysis of the performance of the landsat 8/oli land surface reflectance product. *Remote Sensing of Environment*, 185:46–56, 2016.

---

## Author Response (AR3)

**Authors' Response to Reviews of**

**BAYESIAN ATMOSPHERIC CORRECTION OVER LAND: SENTINEL-2/MSI AND LANDSAT 8/OLI**

Feng Yin, Philip E Lewis, Jose L Gómez-Dans
*Geoscientific Model Development,* `https://doi.org/10.5194/egusphere-2022-170`
* * *
**RC:** *Reviewers' Comment*,     AR: Authors' Response,     ☐ Manuscript Text

**1. Technical corrections**

**RC:** *The title page of \*pdf. manuscript file must include the full institutional addresses of all authors. However, country's and city's names are missing from the affiliation #2. Please add it for the next revision.*

AR:   This has been fixed.

**RC:** *Please ensure that the colour schemes used in your maps and charts allow readers with colour vision deficiencies to correctly interpret your findings. Please check your figures using the Coblis – Color Blindness Simulator (https://www.color-blindness.com/coblis-color-blindness-simulator/) and revise the colour schemes accordingly.*

AR:   File has been checked and colour-blind-safe versions of some figures have been updated.

**RC:** *It would be better to clarify the Bayesian fusion of varying sources of information method.*

AR:   We have added section 2.7 as a clarification/summary:
* * *
**1.1.** **Summary of SIAC approach**

In SIAC, the atmospheric composition at 500m is inferred by combining three sources of constraints:

- an *a priori* constraint on land surface reflectance (at 500m) derived from the MODIS MCD43 product,

- an *a priori* constraint on atmospheric composition (AOT and TCWV) derived from CAMS near-real-time predictions,

- an expectation of spatial smoothness (correlation) in atmospheric composition parameters at the 500m scale.

The spatial and spectral mismatch between the original S2/L8 product and MODIS are dealt with by modelling the MODIS $ePSF$ (Appendix E) and using spectral mapping based on a hyperspectral data library (Appendix D), respectively. These constraints form the observational cost function $\mathcal{J}_{obs}$ (Eq. 2 in Sect. 2.3 ) and prior cost function $\mathcal{J}_{prior}$ (Eq. 5 in Sect. 2.4). By minimizing the combined cost value ($\mathcal{J}_{prior} + \mathcal{J}_{obs}$), the uncertainty-quantified estimates of AOT and TCWV at the 500m resolution
* * *
are obtained. These estimates are then interpolated to the S2/L8 spatial resolution to parameterise Eq. 8 for the atmospheric correction.